# A Computation and Communication Efficient Projection-free Algorithm for Decentralized Constrained Optimization

## Abstract

Decentralized constrained optimization problems arise in numerous real-world applications, where a major challenge lies in the computational complexity of projecting onto complex sets, especially in large-scale systems. The projection-free method, Frank-Wolfe (`FW`), is popular for the constrained optimization problem with complex sets due to its efficiency in tackling the projection process. However, when applying `FW` methods to decentralized constrained finite-sum optimization problems, previous studies provide suboptimal incremental first-order oracle (IFO) bounds in both convex and non-convex settings. In this paper, we propose a stochastic algorithm named Decentralized Variance Reduction Gradient Tracking Frank-Wolfe (`DVRGTFW`), which incorporates the techniques of variance reduction, gradient tracking, and multi-consensus in the `FW` update to obtain tight bounds. We present a novel convergence analysis, diverging from previous decentralized `FW` methods, and demonstrating $\tilde{\mathcal{O}}(n + \sqrt{\frac{n}{m}}L\varepsilon^{-1})$ and $\mathcal{O}(\sqrt{\frac{n}{m}}L^2\varepsilon^{-2})$ IFO complexity bounds in convex and non-convex settings, respectively. To the best of our knowledge, these bounds are the best achieved in the literature to date. Besides, in the non-convex case, `DVRGTFW` achieves $\mathcal{O}(\frac{L^2\varepsilon^{-2}}{\sqrt{1-\lambda_2(W)}})$ communication complexity which is closed to the lower bound $\Omega(\frac{L\varepsilon^{-2}}{\sqrt{1-\lambda_2(W)}})$. Empirical results validate the convergence properties of `DVRGTFW` and highlight its superior performance over other related methods.

## 1 Introduction

Decentralized optimization has gained substantial popularity in applications such as coordinated control, machine learning, and power systems(Latafat et al., 2017; Xin et al., 2020; Dass et al., 2019; Yang et al., 2019). It offers several advantages, including reduced computational burdens for individual agents, enhanced efficiency for system-wide coordination, and the ability to preserve privacy for each participant (Yang et al., 2019; Li et al., 2020b; Xu et al., 2021). Finite-sum optimization problems, which involve minimizing the sum of multiple individual functions, can also benefit from decentralized computation. By distributing the computational effort across multiple agents, decentralized finite-sum optimization could alleviate the computational burden on the central node, which is particularly important for large-scale models (Xin et al., 2022; Hendrikx et al., 2021; Metelev et al., 2024).

In this paper, we focus on the constrained decentralized finite-sum optimization problem with $m$ agents that form a connected network:

$$\min_{x\in\mathcal{X}} \quad f(x) = \frac{1}{m}\sum_{i=1}^{m} f_i(x)$$

$$\text{with} \quad f_i(x) = \frac{1}{n}\sum_{j=1}^{n} f_{i,j}(x), \tag{1}$$

where each agent $i$ has a local objective function $f_i(x)$, composed of $n$ multiple smooth, potentially non-convex functions $f_{i,j}(x)$, and $\mathcal{X} \subset \mathbb{R}^d$ denotes a convex set. The overall objective is to find a

Table 1: Summary of the results on projection free methods for **decentralized stochastic constrained minimization problems**.

| Target case | Method | #IFO | #LMO | #Comm. |
|---|---|---|---|---|
| convex | DeFW (Wai et al., 2017)[1] | $\mathcal{O}(\frac{L\varepsilon^{-1}}{1-\lambda})$ | $\mathcal{O}(\frac{L\varepsilon^{-1}}{1-\lambda})$ | $\mathcal{O}(\frac{L\varepsilon^{-1}}{1-\lambda})$ |
| | DMFW (Hou et al., 2022)[2] | $\mathcal{O}(\frac{L\varepsilon^{-2}}{1-\lambda})$ | $\mathcal{O}(\frac{L\varepsilon^{-2}}{1-\lambda})$ | $\mathcal{O}(\frac{L\varepsilon^{-2}}{1-\lambda})$ |
| | I-PDS (Nguyen et al., 2024) | $\mathcal{O}(L\varepsilon^{-1} + \sigma^2 L\varepsilon^{-2})$ | $\mathcal{O}(L\varepsilon^{-2})$ | $\mathcal{O}(L\varepsilon^{-1})$ |
| | DstoFW (Jiang et al., 2022)[2] | $\mathcal{O}(n^{5/4} + \frac{n^{3/4}L\varepsilon^{-1}}{1-\lambda})$ | $\mathcal{O}(\frac{L\varepsilon^{-1}}{1-\lambda})$ | $\mathcal{O}(\frac{L\varepsilon^{-1}}{1-\lambda})$ |
| | DVRGTFW (Algorithm 1)[2] | $\tilde{\mathcal{O}}(n + \sqrt{\frac{n}{m}}L\varepsilon^{-1})$ | $\tilde{\mathcal{O}}(\sqrt{mn} + L\varepsilon^{-1})$ | $\tilde{\mathcal{O}}(\frac{\sqrt{mn}+L\varepsilon^{-1}}{\sqrt{1-\lambda}})$ |
| non-convex | DeFW (Wai et al., 2017)[1] | $\mathcal{O}(\frac{nL^2\varepsilon^{-2}}{1-\lambda})$ | $\mathcal{O}(\frac{L^2\varepsilon^{-2}}{1-\lambda})$ | $\mathcal{O}(\frac{L^2\varepsilon^{-2}}{1-\lambda})$ |
| | DMFW (Hou et al., 2022)[2] | $\mathcal{O}(\frac{L^2}{(1-\lambda)\exp(\varepsilon^{-1})})$ | $\mathcal{O}(\frac{L^2}{(1-\lambda)\exp(\varepsilon^{-1})})$ | $\mathcal{O}(\frac{L^2}{(1-\lambda)\exp(\varepsilon^{-1})})$ |
| | DstoFW (Jiang et al., 2022)[2] | $\mathcal{O}(n^{4/3} + \frac{n^{2/3}L^2\varepsilon^{-2}}{1-\lambda})$ | $\mathcal{O}(\frac{L^2\varepsilon^{-2}}{1-\lambda})$ | $\mathcal{O}(\frac{L^2\varepsilon^{-2}}{1-\lambda})$ |
| | DVRGTFW (Algorithm 1)[2] | $\tilde{\mathcal{O}}(n + \sqrt{\frac{n}{m}}L\varepsilon^{-2})$ | $\mathcal{O}(L^2\varepsilon^{-2})$ | $\mathcal{O}(\frac{L^2\varepsilon^{-2}}{\sqrt{1-\lambda}})$ |

[1] DeFW is a fully-deterministic algorithm, and the rest are stochastic algorithms.
[2] In fact, their bounds concerning $\lambda$ are worse than those indicated in the table.
*Notation:* $\varepsilon$ = accuracy of the solution, $n$ = size of the dataset assigned to single node, $\sigma^2$ is the variance of the gradient, $L$ = global function's smoothness, $\lambda$ = the second largest eigenvalue of the communication graph, IFO = incremental stochastic first-order oracle, LMO = linear minimization oracle, Comm = Communication.

point $x^*$ that minimizes the average of local functions across $m$ agents within the convex set $\mathcal{X}$. This formation in Eq.(1) plays a crucial role in various real-world applications, especially those requiring large-scale, distributed, and privacy-preserving solutions, such as electric vehicle charging (Zhang et al., 2016) and traffic assignment (Fukushima, 1984).

To solve Eq.(1), the classical approaches, such as the Projection Gradient Descent (PGD) algorithm, are projection-based methods. However, when dealing with complex constraint sets $\mathcal{X}$ or high-dimensional problems, the projection step becomes computationally intensive, making these projection-based methods less efficient and costly (Wai et al., 2017). In contrast, the projection-free methods (i.e., the Frank-Wolfe (FW) algorithm and its variants) address this issue by solving a constrained linear optimization problem instead of performing direct projections (Jaggi, 2013). Wai et al. (2017) propose the first decentralized deterministic FW method based on average consensus. However, this deterministic approach requires a large number of Incremental First-order Oracle (IFO) calls, which significantly increases computational costs. Consequently, subsequent research has focused on developing stochastic decentralized FW methods to reduce the number of IFO calls. For instance, Gao et al. (2021); Xie et al. (2019) propose decentralized FW methods, incorporating variance reduction techniques, for the DR-submodular optimization problem to reduce computation overhead. Nguyen et al. (2024) propose a communication-efficient decentralized FW method by combining with the conditional gradient sliding technique (Lan & Zhou, 2016). Hou et al. (2022) utilizes the momentum technique (Nesterov, 1983) to improve the convergence rate for the decentralized stochastic FW method. (Jiang et al., 2022) also adopt the variance reduction technique to develop computation and communication efficient decentralized FW method for both convex and nonconvex optimization problems. Notably, compared to the best IFO bounds in centralized settings (Beznosikov et al., 2024) ($\tilde{\mathcal{O}}(n + \frac{1}{\varepsilon})$ and $\tilde{\mathcal{O}}(n + \frac{\sqrt{n}}{\varepsilon^2})$ for convex and non-convex optimization, respectively), the current decentralized FW methods achieve the suboptimal IFO complexity. We summarize representative decentralized FW methods and their key characteristics in Table 1.

In this paper, we focus on developing a decentralized stochastic Frank-Wolfe algorithm that is both computationally and communication efficient, aiming to minimize the computational and communication overhead in decentralized settings. Inspired by the existing loopless variance reduction technique (Li et al., 2021; Beznosikov et al., 2024) and decentralized optimization methods (Wai et al., 2017; Pu & Nedić, 2021), we propose a decentralized variance reduction gradient tracking method (DVRGTFW) to solve Eq.(1). We present a different proof compared to Wai et al. (2017); Jiang et al. (2022), and demonstrate the best rates of DVRGTFW in both convex and non-convex settings. The contributions of this paper are summarized as follows:

- **The best-known IFO complexity both in the convex case and the non-convex case.**
  For convex case, DVRGTFW achieves an improved IFO complexity of $\tilde{\mathcal{O}}(n + \sqrt{\frac{n}{m}}L\varepsilon^{-1})$, which represents a significant advancement compared to the decentralized stochastic meth-

ods proposed by (Jiang et al., 2022) with respect to the dataset size $n$. Furthermore, the theoretical convergence rates of our method outperform those reported in (Hou et al., 2022; Nguyen et al., 2024) in terms of the accuracy $\varepsilon$. For non-convex case, `DVRGTFW` attains an improved IFO complexity of $\mathcal{O}(\sqrt{\frac{n}{m}}L^2\varepsilon^{-2})$. This significantly improves the result reported in (Jiang et al., 2022) with respect to the dataset size $n$. Additionally, the theoretical convergence rates of our methodology surpass those reported in (Hou et al., 2022; Nguyen et al., 2024) in terms of the accuracy $\varepsilon$. When the number of nodes $m$ is set to 1, both results align with the optimal outcome reported in (Beznosikov et al., 2024).

- **Nearly optimal in non-convex communication complexity.** For non-convex case, `DVRGTFW` has the first near-optimal communication perplexity $\mathcal{O}(\frac{L^2\varepsilon^{-2}}{\sqrt{1-\lambda_2(W)}})$, where $\lambda_2(W)$ is the second-largest eigenvalues of the gossip matrix $W$. This result is close to the lower bound of communication complexity (Lu & De Sa, 2021), which is $\Omega(\frac{L\varepsilon^{-2}}{\sqrt{1-\lambda_2(W)}})$ for finding an $\varepsilon$-stationary point of smooth non-convex function via a first-order algorithm.

## 2 RELATED WORK

Below we provide a review of related literature that shapes our study.

**Variance Reduction** Variance reduction techniques leverage the control variate technique (Rubinstein & Marcus, 1985) to reduce inherent sampling variance in stochastic methods, thereby achieving the same convergence rate as the deterministic methods. The classic SVRG method (Johnson & Zhang, 2013) adopts a double-loop structure, maintaining a snapshot of model parameters to compute the full gradient in the outer loop and constructing an unbiased gradient estimate in the inner loop. Moreover, Nguyen et al. (2017); Fang et al. (2018) admits a simple recursive framework and demonstrates the best IFO complexity for non-convex optimization problems. Besides, Li et al. (2021) proposes a novel and practical loopless variance-reduced technique.

**Gradient Tracking** In the decentralized setting, the heterogeneity in agents' local data distributions increases the communication cost. To enhance communication efficiency, Nedic et al. (2017); Pu & Nedić (2021); Qu & Li (2020) propose the gradient tracking technique. This technique achieves communication efficiency by maintaining the accuracy of first-order information through tracking the average of local gradients. Moreover, Ye et al. (2023a) demonstrated that combining gradient tracking with multi-consensus (Arioli & Scott, 2014; Li et al., 2020a) makes the analysis of decentralized algorithms closer to their centralized counterparts, making it particularly useful for decentralized convex optimization.

**Variance Reduction in Frank-Wolfe** Building upon variance reduction techniques, an increasing number of centralized stochastic `FW`-type methods have been proposed to address the variance introduced by stochastic gradients (e.g., (Hazan & Kale, 2012; Hazan & Luo, 2016; Reddi et al., 2016; Yurtsever et al., 2019; Weber & Sra, 2022; Beznosikov et al., 2024)). For convex finite-sum optimization, to achieve an $\varepsilon$-solution, Beznosikov et al. (2024) combined a stochastic recursive gradient technique (Nguyen et al., 2017) with the classical Frank-Wolfe algorithm to achieve the best-known IFO complexity $\tilde{\mathcal{O}}(n + \frac{\sqrt{n}}{\varepsilon})$ and LMO complexity $\tilde{\mathcal{O}}(\sqrt{n} + \frac{1}{\varepsilon})$. For non-convex finite-sum optimization, Yurtsever et al. (2019) utilized a stochastic path integrated differential estimator technique (Fang et al., 2018) with the classical `FW` method to attain the best IFO and LMO complexity both at $\mathcal{O}(\frac{\sqrt{n}}{\varepsilon^2})$, matching the result of Beznosikov et al. (2024).

## 3 NOTATION AND ASSUMPTIONS

Let $\langle x, y \rangle = \sum_{i=1}^{d} x_i y_i$ denote the standard inner product of vectors $x, y \in \mathbb{R}^d$, with this notation we can introduce the standard $l_2$-norm in $\mathbb{R}^d$ in the following way: $\|x\| = \sqrt{\langle x, x \rangle}$. The notation $[m]$ is the abbreviation of the set $\{1, \ldots, m\}$. $\mathbf{1}$ denotes a column vector with all elements of 1.

Moreover, we define aggregate variables for all agents as

$$\mathbf{x} = \begin{bmatrix} \mathbf{x}_1^\top \\ \vdots \\ \mathbf{x}_m^\top \end{bmatrix} \in \mathbb{R}^{m \times d},$$

where each $\mathbf{x}_i \in \mathbb{R}^d$ are the local variable on the $i$-th agent. We use the lower case with the bar to represent the mean vector, such that $\bar{x} = \frac{1}{m} \sum_{i=1}^{m} x_i \in \mathbb{R}^d$. Furthermore, we define the matrix of aggregate gradients as

$$\nabla \mathbf{f}(\mathbf{x}) = \begin{bmatrix} \nabla f_1(\mathbf{x}_1)^\top \\ \vdots \\ \nabla f_m(\mathbf{x}_m)^\top \end{bmatrix} \in \mathbb{R}^{m \times d}.$$

Then we introduce the following assumptions on the constrained decentralized finite-sum optimization problem 1

**Assumption 1** *The global function $f$ is convex. i.e., for any $x, y \in \mathcal{X}$,*

$$f(x) \geq f(y) + \langle \nabla f(y), x - y \rangle.$$

**Assumption 2** *The individual function $\{f_{i,j}\}_{j=1}^n$ on each agent are $L$-average smooth for some $L \geq 0$. i.e., for any $x, y \in \mathcal{X}$,*

$$\frac{1}{n} \sum_{j=1}^{n} \|\nabla f_{i,j}(x) - \nabla f_{i,j}(y)\|^2 \leq L^2 \|x - y\|^2,$$

*in addition, the global function $f$ is bounded below, i.e., $f^* = \inf_{\mathbf{x} \in \mathbb{R}^d} f(\mathbf{x}) > -\infty$.*

**Assumption 3** *The set $\mathcal{X}$ is convex and compact with a diameter $D$, i.e., for any $x, y \in \mathcal{X}$,*

$$\|x - y\| \leq D.$$

Note we consider both convex and non-convex cases of the global function $f$, but even if $f$ is convex, we do not additionally assume that each individual function is convex, hence, it can be used in a wider range of applications, for example, the sub-problem of Fast PCA (Gang & Bajwa, 2022) by the shift-invert method is non-convex. Assumption 2 and Assumption 3 are standard in the optimization literature and widely used in the analysis of Frank-Wolfe-type methods.

For decentralized optimization, we use the gossip matrix $W \in \mathbb{R}^{m \times m}$ to characterize the behavior of agents updating local variables by the weighted sum of information from the neighbors. Moreover, we use $\lambda_2(\mathbf{W})$ to denote its second largest singular value, and we assume the matrix $W$ satisfies

**Assumption 4** *The gossip matrix $W \in [0,1]^{n \times n}$ is doubly stochastic, that is $W\mathbf{1} = \mathbf{1}$, and $\mathbf{1}^\top W = \mathbf{1}^\top$.*

## 4 METHOD

Based on the classic Decentralized Frank-Wolfe Algorithm (Wai et al. (2017)) and `SARAH` (specifically the loopless version (Li et al. (2021))), we propose Decentralized Variance Reduction Frank-Wolfe Algorithm named `DVRGTFW`, as outlined in Algorithm 1. The centralized `FW` algorithm for constrained problem can be proceeded by the following iteration:

$$\mathbf{d}_t = \arg\min_{d \in \mathcal{X}} \langle \nabla f(\mathbf{x}_t), \mathbf{d} \rangle, \tag{2a}$$

$$\mathbf{x}_{t+1} = \mathbf{x}_t + \eta_t(\mathbf{d}_t - \mathbf{x}_t), \tag{2b}$$

where $\eta_t \in (0, 1]$ is a step size to be determined. Given that $\mathbf{x}_{t+1}$ is a convex combination of $\mathbf{x}_t$ and $\mathbf{d}_t$, it follows that $\mathbf{x}_{t+1}$ also lies in the convex set $\mathcal{X}$. We note that the linear optimization in Eq.(2a) can be solved more efficiently than the projection operation. In the decentralized setting, our method (i.e., `DVRGTFW`) follows the spirit and avoids the complex projection operation by having

each agent perform a linear minimization over the constraint set $\mathcal{X}$. Each agent then takes a convex combination of the optimal $\mathbf{d}_{i,t}$ and $\mathbf{x}_{i,t}$. Finally, agents communicate with neighbours to update and obtain a feasible variable estimate $\mathbf{x}_{i,t+1}$ within the constraint set $\mathcal{X}$.

---

**Algorithm 1** Decentralized variance reduction gradient tracking Frank-Wolfe (DVRGTFW)

---

1: **Input**:initial parameter $\bar{x}_0 \in \mathbb{R}^d$, step size $\{\eta_t\}_{t \geq 0}$,
    probability $p \in (0, 1]$, mini-batch size b, numbers of communication rounds $K_{in}$ and $K$.
2: $\mathbf{x}_0 = \mathbf{1}\bar{x}_0, \mathbf{v}_0 = \nabla \mathbf{f}(\mathbf{x}_0)$
3: $\mathbf{y}_0 = \textbf{FastMix}(\mathbf{v}_0, K_{in})$
4: **for** $t = 0, \ldots, T - 1$ **do**
5:     $\gamma_t \sim \text{Bernoulli}(p)$
6:     $\mathbf{d}_t = \arg \min_{\mathbf{d} \in \mathcal{X}} \langle \mathbf{y}_t, d \rangle$
7:     $\mathbf{x}_{t+1} = \textbf{FastMix}(\mathbf{x}_t + \eta_t(\mathbf{d}_t - \mathbf{x}_t), K)$
8:     **parallel for** $i = 1, \ldots, n$ **do** $v_{i,t}$

9:     $\mathbf{v}_{i,t+1} = \begin{cases} \nabla f_i(\mathbf{x}_{i,t+1}), & \text{if } \gamma_t = 1, \\ \mathbf{v}_{i,t} + \dfrac{1}{b} \sum_{j=1}^{b} \left( \nabla f_{i,\xi_j}(\mathbf{x}_{i,t+1}) - \nabla f_{i,\xi_j}(\mathbf{x}_{i,t}) \right), & \text{otherwise}, \end{cases}$

       where each $\xi_{i,\xi_j}$ is uniformly and independently sampled from $\{1, \ldots, n\}$
10:     **end parallel for**
11:     $\mathbf{y}_{t+1} = \textbf{FastMix}(\mathbf{y}_t + \mathbf{v}_{t+1} - \mathbf{v}_t, K)$
12: **end for**

---

**Algorithm 2** FastMix$(\mathbf{u}^{(0)}, K)$

---

1: **Initialize:** $\mathbf{u}^{(-1)} = \mathbf{u}^{(0)}, \eta_u = \frac{1 - \sqrt{1 - \lambda_2^2(W)}}{1 + \sqrt{1 - \lambda_2^2(W)}}$.
2: **for** $k = 0, 1, \ldots, K$ **do**
3:     $\mathbf{u}^{(k+1)} = (1 + \eta_u)W\mathbf{u}^{(k)} - \eta_u\mathbf{u}^{(k-1)}$
4: **end for**

---

To accelerate the decaying rate of consensus error, we use the subroutine `FastMix` (Algorithm 2) and gradient tracking, `FastMix` can help variable communicate with neighbours faster, and gradient tracking step can take advantage of the gradient information from the last step to estimate the gradient of global function $f$, so the update of local variables can be written as

$$\begin{cases} \mathbf{x}_{t+1} = \textbf{FastMix}(\mathbf{x}_t + \eta_t(\mathbf{d}_t - \mathbf{x}_t), K), \\ \mathbf{y}_{t+1} = \textbf{FastMix}(\mathbf{y}_t + \mathbf{v}_{t+1} - \mathbf{v}_t, K). \end{cases} \tag{3}$$

Lemma 2 in Appendix A demonstrates that $\bar{x}_{t+1}$ can be interpreted as a convex combination of $\bar{x}_t$ and $\bar{d}_t$. Furthermore, Lemma 3 in the same Appendix indicates that each $\mathbf{x}_{i,t}$ and $\mathbf{v}_{i,t}$ is approximately close to $\bar{x}_t$ and $\bar{v}_t$ respectively, and with an increase in the number of communications, the consensus error is expected to decrease.

To address the variance on gradient caused by random samples, we use a kind of variance-reduced method named `SARAH` (Nguyen et al., 2017) which changes the deterministic gradient in the conditional gradient method to some stochastic gradient $\mathbf{v}_{t+1}$ as:

$$\mathbf{v}_{t+1} = \begin{cases} \nabla f_i(\mathbf{x}_{t+1}), & \text{with probability } p, \\ \mathbf{v}_t + \dfrac{1}{b} \sum_{i \in S_k} \left( \nabla f_i(\mathbf{x}_{t+1}) - \nabla f_i(\mathbf{x}_t) \right), & \text{with probability } 1 - p, \end{cases} \tag{4}$$

where $S_k$ is a random batch sampled from dataset with size $b$, as noted in the original paper on `SARAH`, this method has better convergence guarantees and smoother convergence paths with fewer oscillations than `SVRG`, making `SARAH` preferred in both theory and practice. As a result, the

construction of $\mathbf{v}_{i,t}$ follows the probabilistic recursive way like (Li et al., 2021) which is more general for `DVRGTFW` to switch between the exact gradient and batch gradient.

**Remark 1** *The multi-consensus step in our algorithm can analog the decentralized `SARAH` Frank-Wolfe algorithm more efficiently and leads the convergence analysis to be the same as standard analysis (Beznosikov et al., 2024), In contrast, `DstoFW` (Jiang et al., 2022) does not have such a good property and it can not achieve near-optimal computation complexity nor near-optimal communication complexity.*

## 5 CONVERGENCE ANALYSIS

### 5.1 CONVEX CASE

First, we give the convergence of `DVRGTFW` in the convex case.

**Theorem 1** *Under Assumption 1, 2, 3 and 4, we run `DVRGTFW` with*

$$b = \left\lceil 3\sqrt{\frac{2n}{m}} \right\rceil, \quad p = \frac{2b}{n+2b}, \quad K_{in} = \left\lceil \frac{\log\left(\|\mathbf{v}_0 - \mathbf{1}\bar{v}_0\|^2 / L^2\right)}{\sqrt{1 - \lambda_2(W)}} \right\rceil, \quad K = \left\lceil \frac{3}{\sqrt{1 - \lambda_2(W)}} \right\rceil,$$

*and for any $T$ one can choose $\{\eta_t\}_{t \geq 0}$ as follows:*

$$\begin{aligned}
&\text{if } T \leq \frac{2}{p}, && \eta_t = \frac{p}{2}, \\
&\text{if } T > \frac{2}{p} \text{ and } t < \left\lceil \frac{T}{2} \right\rceil, && \eta_t = \frac{p}{2}, \\
&\text{if } T > \frac{2}{p} \text{ and } t \geq \left\lceil \frac{T}{2} \right\rceil, && \eta_t = \frac{2}{(4/p + t - \lceil T/2 \rceil)},
\end{aligned}$$

*For the setting of $b$, $p$, $K_{in}$, $K$ and the choice of $\eta_t$, we have the following convergence:*

$$\mathbb{E}[f(\bar{x}^T) - f(x^*)] = \mathcal{O}\left( \frac{f(\bar{x}^0) - f(x^*) + \frac{1}{6}}{p} \exp\left(-\frac{pT}{4}\right) + \frac{8LD^2}{T} \right).$$

The complete proof is provided in Appendix C. Since `DVRGTFW` estimates the gradient recursively by using the mini-batch gradient with high probability $1 - p$ and computing the exact gradient with low probability $p$, one can note that for each iteration, we on average compute the stochastic gradient $(pn + (1-p) \cdot 2b) * m$ times. If we take $p$ close to 1, the guarantees in Theorem 1 gives faster convergence, but the oracle complexity per iteration increases. For instance, if we take $p = 1$, we simply obtain a deterministic method, and the estimates for convergence and the number of gradient calculations reproduce the results for the classic decentralized Frank-Wolfe algorithm, on the other hand, if we take $p = 0$, the number of stochastic gradient calls per iteration decreases, but the iterative convergence rate drops. It is optimal to choose $p$ based on the condition: $pn = 2(1-p)b$, $i.e. p = \frac{2b}{n+2b}$, also it is optimal to set $b = \mathcal{O}(\sqrt{\frac{n}{m}})$ and set the step size $\eta_t$ as above. Note that each agent need to use the same seed to generate the Bernoulli distributed variable $\gamma_t$, which enforces all agents always share the identical $\gamma_t$. Then we show that under the above settings, we can obtain the following result.

**Corollary 1** *Under the conditions of Theorem 1, for each node $i \in [m]$, `DVRGTFW` achieves an $\epsilon$ suboptimality with*

$$\mathcal{O}(\sqrt{mn} \log \frac{1}{\epsilon} + \frac{LD^2}{\epsilon}) \quad \textit{LMO calls,}$$

$$\mathcal{O}(n \log \frac{1}{\epsilon} + \sqrt{\frac{n}{m}} \frac{LD^2}{\epsilon}) \quad \textit{IFO calls, and}$$

$$\mathcal{O}(\frac{\sqrt{mn} \log \frac{1}{\epsilon} + \frac{LD^2}{\epsilon}}{\sqrt{1 - \lambda_2(W)}}) \quad \textit{rounds of communication.}$$

Under the setting in Theorem 1, the required number of the stochastic gradient computations is $\mathcal{O}(n \log \frac{1}{\epsilon} + \sqrt{\frac{n}{m}} \frac{LD^2}{\epsilon})$, and the LMO complexity is $\mathcal{O}(\sqrt{mn} \log \frac{1}{\epsilon} + \frac{LD^2}{\epsilon})$, when reduce to centralized setting ($m = 1$), the result match the optimal result in Beznosikov et al. (2024), and the communication rounds $K$ at each iteration is deterministic which equals to $\left\lceil \frac{3}{\sqrt{1-\lambda_2(W)}} \right\rceil$, and combining with the initial communication rounds $K_{in} = \left\lceil \frac{\log(\|\mathbf{v}_0 - \mathbf{1}\bar{v}_0\|^2/L^2)}{\sqrt{1-\lambda_2(W)}} \right\rceil$, the total communication complexity is $\mathcal{O}(\frac{\sqrt{mn} \log \frac{1}{\epsilon} + \frac{LD^2}{\epsilon}}{\sqrt{1-\lambda_2(W)}})$ in expectation.

**Remark 2** *According to the setting in Theorem 1, increasing the batch size $b$ leads to a higher probability $p$ of obtaining the full gradient. Guaranteed by Theorem 1, this enhancement results in faster convergence, thereby reducing communication costs. However, it also leads to an increase in the oracle complexity per iteration. Therefore, it is essential to select an appropriate batch size $b$ to balance computational complexity with communication complexity.*

## 5.2 NON-CONVEX CASE

Then we give the convergence of `DVRGTFW` in the non-convex case. Note that in the centralized setting, Jaggi (2013) gives the *Frank-Wolfe gap* function as a criterion for convergence:

$$\boldsymbol{gap}(y) = \max_{x\in\mathcal{X}}\langle \nabla f(y), y - x \rangle,$$

Lacoste-Julien (2016) notes that the Frank-Wolfe gap is a meaningful measure of non-stationarity and serves as an affine-invariant generalization of the more standard convergence criterion $\|\nabla f(y)\|$ which is used for unconstrained non-convex problems. In the decentralized setting, *Frank-Wolfe gap* is slightly modified which is defined as follows:

$$\boldsymbol{gap}(\bar{x}^t) = \max_{x\in\mathcal{X}}\langle \nabla f(\bar{x}^t), \bar{x}^t - x \rangle,$$

from the definition, when $\boldsymbol{gap}(\bar{x}^t) = 0$, the iterate $\bar{x}^t$ will be a stationary point to Eq.(1), thus we regard $\boldsymbol{gap}(\bar{x}^t)$ as a measure of the stationarity of the iterate $\bar{x}^t$. Follow the assumption in (Wai et al., 2017), we define the set of stationary point to (1) as:

$$\mathcal{X}^\star = \left\{ \underline{x} \in \mathcal{X} : \max_{x\in\mathcal{X}}\langle \nabla F(\underline{x}), \underline{x} - x \rangle = 0 \right\}.$$

We consider the following technical assumption:

**Assumption 5** *The set $\mathcal{X}^\star$ is non-empty. Moreover, the function $f(x)$ takes a finite number of values over $\mathcal{X}^\star$, i.e., the set $f(\mathcal{X}^\star) = \{f(x) : x \in \mathcal{X}^\star\}$ is finite.*

It is reasonable to assume that Eq.(1) has a finite number of stationary points since the set $\mathcal{X}$ is bounded, thus Assumption 5 is satisfied. Then the following theorem is valid.

**Theorem 2** *Under Assumption 2, 3 and 4, we run `DVRGTFW` with*

$$b = \left\lceil 3\sqrt{\frac{n}{2m}} \right\rceil, \quad p = \frac{2b}{2b+n}, \quad K_{in} = \left\lceil \frac{\log\left(\|\mathbf{v}_0 - \mathbf{1}\bar{v}_0\|^2/L^2\right)}{\sqrt{1-\lambda_2(W)}} \right\rceil, \quad K = \left\lceil \frac{3}{\sqrt{1-\lambda_2(W)}} \right\rceil,$$

*and if we set $\eta_t = \frac{1}{\sqrt{T}}$, we have the following convergence:*

$$\mathbb{E}\left[\min_{0\le t\le T-1} \boldsymbol{gap}(\bar{x}^t)\right] = \mathcal{O}(\frac{f(\bar{x}^0) - f(x^*) + \frac{2\sqrt{2}}{7}}{\sqrt{T}} + \frac{7LD^2}{\sqrt{T}}).$$

The proof can be found in Appendix D. When the set $\mathcal{X}^\star$ satisfy Assumption 5, like proof in Wai et al. (2017), according to we can apply Nurminskii's sufficient condition (Theorem 1 from Zangwill (1969)) to prove that for `DVRGTFW`, every limit point of $\{\bar{x}^t\}_{t\ge 1}$ belongs to $\mathcal{X}^*$. Similar to the analyse in convex case, under the setting of $p, b, \eta_t$ in Theorem 2, we now have

**Corollary 2** *Under the conditions of Theorem 2, for each node $i \in [n]$, DVRGTFW achieves an $\epsilon$ suboptimality with*

$$\mathcal{O}\left(\left[\frac{g_0}{\epsilon}\right]^2 + \left[\frac{LD^2}{\epsilon}\right]^2\right) \quad \text{LMO calls,}$$

$$\mathcal{O}\left(\sqrt{\frac{n}{m}}\left[\frac{g_0}{\epsilon}\right]^2 + \sqrt{\frac{n}{m}}\left[\frac{LD^2}{\epsilon}\right]^2\right) \quad \text{IFO calls, and}$$

$$\mathcal{O}\left(\frac{\left[\frac{g_0}{\epsilon}\right]^2 + \left[\frac{LD^2}{\epsilon}\right]^2}{\sqrt{1 - \lambda_2(W)}}\right) \quad \text{rounds of communication,}$$

*where $g_0 = f(\bar{x}^0) - f(x^*) + \frac{2\sqrt{2}}{7}$.*

Under the setting in Theorem 2, the required number of the stochastic gradient computations is $\mathcal{O}\left(\sqrt{\frac{n}{m}}\left[\frac{g_0}{\epsilon}\right]^2 + \sqrt{\frac{n}{m}}\left[\frac{LD^2}{\epsilon}\right]^2\right)$, and the LMO complexity is $\mathcal{O}\left(\left[\frac{g_0}{\epsilon}\right]^2 + \left[\frac{LD^2}{\epsilon}\right]^2\right)$, when reduce to centralized setting ($m = 1$), the result match the optimal result in (Beznosikov et al., 2024), and the communication rounds $K$ at each iteration is deterministic which equals to $\left\lceil \frac{3}{\sqrt{1-\lambda_2(W)}} \right\rceil$, and combining with the initial communication rounds $K_{in} = \left\lceil \frac{\log\left(\|\mathbf{v}_0 - \mathbf{1}\bar{v}_0\|^2/L^2\right)}{\sqrt{1-\lambda_2(W)}} \right\rceil$, the total communication complexity is $\mathcal{O}\left(\frac{\left[\frac{g_0}{\epsilon}\right]^2 + \left[\frac{LD^2}{\epsilon}\right]^2}{\sqrt{1-\lambda_2(W)}}\right)$ in expectation.

**Remark 3** *Corollary 1 and Corollary 2 shows that FastMix can eliminate IFO complexity's dependence on $\lambda_2(W)$, which means that the structure of the communication graph will not influence the IFO complexity and LMO complexity. Compared with the existing decentralized Frank-Wolfe algorithms, our algorithm obtain the optimal bound in IFO and LMO complexity, and for non-convex finute-sum problem, our communication bound $\mathcal{O}\left(\frac{L^2\varepsilon^{-2}}{\sqrt{1-\lambda_2(W)}}\right)$ is nearly optimal to the lower bound $\mathcal{O}\left(\frac{L\varepsilon^{-2}}{\sqrt{1-\lambda_2(W)}}\right)$ in (Corollary 1, Lu & De Sa (2021)).*

## 6 EXPERIMENT

We evaluate the performance of our algorithms on logistic regression with different settings, including the situation in which each $f_i(x)$ is convex and the local function $f_i(x)$ is non-convex. In our experiment, the constrained set is set as an $l_1$ norm ball constraint $\omega = \{x | \|x\|_1 \leq R\}$, for simplicity, we constantly take $R = 20$ of the constrained set in the following experiments.

### 6.1 EXPERIMENT SETTINGS

#### 6.1.1 THE SETTING OF NETWORKS

In our experiments, we consider random networks where each pair of agents has a connection with a probability of $p$. We set $W = I - L/\lambda_1(L)$, where $L$ is the Laplacian matrix associated with a weighted graph, and $\lambda_1(L)$ is the largest eigenvalue of $L$. We also set the number of agents as $n = 100$. In our experiments, we run the algorithms on the setting of $p = 0.1$ and $p = 0.5$, which correspond to $1 - \lambda_2(W) = 0.05$ and $1 - \lambda_2(W) = 0.81$ respectively.

#### 6.1.2 THE CHOICE OF DATASET

We conduct our experiments on two real-world binary classification datasets from LIBSVM data repository[1], one of the two datasets we deliberately selected have more data points and fewer features, leading to high computation complexity, while the other has relatively fewer data points but more features, making it more challenging to converge. We summarize it in Table 2.

---

[1] https://www.csie.ntu.edu.tw/~cjlin/libsvmtools/datasets/binary.html

Table 2: Real Datasets For Binary Classification

| Dataset | $d$ | $mn$ |
|---|---|---|
| real-sim | 20959 | 72309 |
| covtype.binary | 54 | 581014 |

## 6.2 EXPERIMENTS ON CONVEX LOGISTIC REGRESSION

We consider the convex logistic regression model in which the local objective function of logistic regression is defined as

$$f_i(x) = \frac{1}{n}\sum_{j=1}^{n}\log(1 + \exp(-l_{i,j}\langle a_{i,j}, x\rangle)), \tag{5}$$

where $a_{i,j} \in \mathbb{R}^d$ is the feature vector of the $j$th local sample of agent $i$, $l_{i,j} \in \{-1, 1\}$ is the classification value of the $j$th local sample of agent $i$, we compare our algorithm (DVRGTFW) with Decentralized Frank-Wolfe algorithm (DeFW) in Wai et al. (2017) and Decentralized Spider Frank-Wolfe algorithm (DstoFW) in Jiang et al. (2022), The parameters of all algorithms are well-tuned to achieve their best performances, and we set the batch size $b$ as $\mathcal{O}(\sqrt{\frac{n}{m}})$ level and number of communications per step $K$ as $\mathcal{O}(1)$ level in DVRGTFW, note that in the experiment, we do not use a extreme graph structure with a significantly big $\lambda_2(W)$, so such communication setting accord with our theoretical analyze. Futhermore, we initialize $\mathbf{x}_0 = \mathbf{0}$ for all the compared methods. In a convex setting, we report the experimental results in Figure 1.

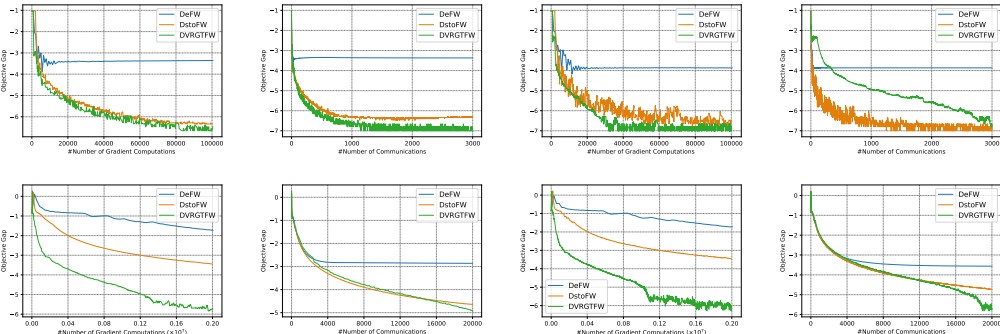

Figure 1: Comparisons with convex logistic regression and random networks. Each local objective $f_i(x)$ may be non-convex. In the top row, experiments on real-sim dataset for the agent $i = 1\ldots, m$. In the bottom row, experiments on covtype.binary dataset for the agent $i = 1\ldots, m$. Random networks have $1 - \lambda_2(W) = 0.05$ in the left two columns and $1 - \lambda_2(W) = 0.81$ in the right two columns. **Objective Gap** is defined as $f(\bar{x}_t) - f(x^*)$, where $f(x^*)$ is obtained by the PGD algorithm (Bubeck et al., 2015).

Compared to DsgFW, DVRGTFW demonstrates superior computational efficiency across both datasets, irrespective of the random graph's structure. This advantage is particularly evident in the covtype.binary dataset, which contains a larger number of data points, aligning well with the theoretical computational complexity results of our algorithm. Moreover, our algorithm almost achieves lower communication costs than both DstoFW and DeFW in all cases.

## 6.3 EXPERIMENTS ON NONCONVEX LOGISTIC REGRESSION

We consider the non-convex logistic regression model in which the local objective function of logistic regression is defined as

$$f_i(x) = \frac{1}{n}\sum_{j=1}^{n}\frac{1}{1 + \exp(l_{i,j}\langle a_{i,j}, x\rangle)}, \tag{6}$$

where $a_{i,j}$ and $l_{i,j}$ are same as those in Eq.(5). The step size of DeFW, DstoFW and DVRGTFW are $\frac{1}{\sqrt{t}}$. As same as the convex setting, the parameters of all algorithms are well-tuned to achieve

their best performances, and in `DVRGTFW`, we set batch size $b$ as $\mathcal{O}(\sqrt{\frac{m}{n}})$ level and number of communications $K$ as $\mathcal{O}(1)$ level. Moreover,we initialize $\mathbf{x}_0 = \mathbf{0}$ for all the compared methods. In a non-convex setting, we report the experimental results in Figure 2.

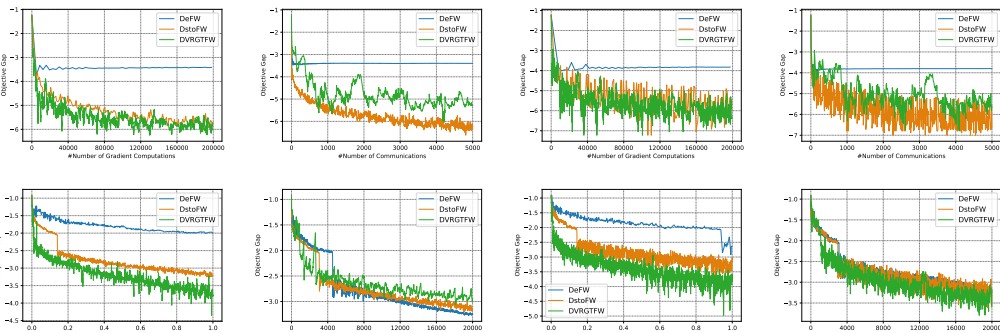

Figure 2: Comparisons with non-convex logistic regression and random networks. Each local objective $f_i(x)$ may be non-convex. In the top row, experiments on `real-sim` dataset for the agent $i = 1 \ldots, m$. In the bottom row, experiments on `covtype.binary` dataset for the agent $i = 1 \ldots, m$. Random networks have $1 - \lambda_2(W) = 0.05$ in the left two columns and $1 - \lambda_2(W) = 0.81$ in the right two columns. **Objective Gap** is defined as $f(\bar{x}_t) - f(x_{min})$, where $f(x_{min})$ is the minimum value of the function obtained from multiple runs of `PGD`.

From Figure 2, it is clear that our algorithm demonstrates superior computational complexity for the non-convex problem, aligning well with our theoretical findings. However, our algorithm performs significantly worse than the `DstoFW` algorithm on the real-sim dataset, which contradicts our theoretical expectations. Perhaps the large number of features in the real-sim dataset makes it challenging for the Frank-Wolfe algorithm to converge. Additionally, the bounds analyzed by the compared algorithms might not be sufficiently tight. Nonetheless, by increasing the batch size, comparable results can be achieved. Thus, for non-convex problems, adjusting the batch size allows us to balance communication complexity and computation complexity. It is important to note that this adjustment is made at a constant level.

## 7 CONCLUSION

In this paper, we propose `DVRGTFW`, a novel decentralized projection-free algorithm tailored for constrained decentralized finite-sum optimization problems. Compared to existing decentralized stochastic projection-free algorithms, our method eliminates the need for large batch computations, thereby improving efficiency. Notably, `DVRGTFW` achieves the best-known IFO complexity for both convex and non-convex scenarios, and it effectively reduces communication complexity to approach theoretical lower bounds for non-convex problems. Besides, it can reduce to the optimal result in a centralized setting. Comprehensive numerical experiments validate our theoretical analysis and demonstrate the practical effectiveness of `DVRGTFW`.

The design of `DVRGTFW` is grounded in an innovative framework that integrates loopless variance-reduced iteration, gradient tracking, and multi-consensus techniques. The proof of `DVRGTFW` utilizes a Lyapunov function that captures the function value, global and local gradient estimation errors, and consensus errors, yielding an intuitive and easy-to-follow analysis framework.

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

# A  TECHNICAL LEMMAS

In this section, we will introduce several useful lemmas that will be used in our proofs. They are easy to check or prove, so we omit the details of these lemmas.

**Lemma 1** *For any $x_1, \ldots, x_N \in \mathbb{R}^d$ in the following inequality holds:*

$$\left\| \sum_{i=1}^m x_i \right\|^2 \leq N \sum_{i=1}^m \|x_i\|^2.$$

**Lemma 2** *(Lemma 2 from Ye et al. (2023b)). For Frank-Wolfe update in `DVRGTFW`, we have $\bar{y}_t = \bar{v}_t$.*

**Lemma 3** *(Lemma 2 from Liu & Morse (2011)). Under Assumption 4, `FastMix` holds that*

$$\frac{1}{n}\mathbf{1}^\top \mathbf{u}^K = \bar{u}^0 \quad and \quad \left\| \mathbf{u}^K - \mathbf{1}^\top u^0 \right\| \leq \left( 1 - \sqrt{1 - \lambda_2(W)} \right)^K \left\| \mathbf{u}^0 - \mathbf{1}\bar{u}^0 \right\|,$$

*where $\bar{u}^0 = \frac{1}{n}\mathbf{1}^\top \mathbf{u}^0$.*

**Lemma 4** *(Lemma 3 from Ye et al. (2023b)). For any $\mathbf{s} \in \mathbb{R}^{n \times d}$, we have $\|\mathbf{s} - \mathbf{1}\bar{s}\| \leq \|\mathbf{s}\|$, where $\bar{s} = \frac{1}{m}\mathbf{1}^\top \mathbf{s}$*

**Lemma 5** *(Lemma 4 from Luo & Ye (2022)).Under Assumption 2, we have $\|\nabla \mathbf{f}(x) - \nabla \mathbf{f}(y)\| \leq L\|x - y\|$ for any $x, y \in \mathbb{R}^{n \times d}$*

**Lemma 6** *(Lemma 1.2.3 from Nesterov (2013)). Suppose that $f$ is $L$-smooth. Then, for any $x, y \in \mathbb{R}^d$,*

$$f(x) \leq f(y) + \langle \nabla f(y), x - y \rangle + \frac{L}{2}\|x - y\|^2$$

**Lemma 7** *(Lemma 3 from Stich (2019)). Let $\{r_k\}_{k \geq 0}$ is a non-negative sequence, which satisfies the relation*

$$r_{k+1} \leq (1 - \eta_k)r_k + c\eta_k^2.$$

*Then there exists stepsizes $\eta_k \leq \frac{1}{d}$, such that:*

$$r_K = \mathcal{O}\left(dr_0 \exp\left(-\frac{K}{2d}\right) + \frac{c}{K}\right).$$

*In particular, the step size are chosen as follows:*

$$\text{if} \quad K \leq d, \qquad\qquad \eta_k = \frac{1}{d},$$

$$\text{if} \quad K > d \quad \text{and} \quad k < k_0, \qquad\qquad \eta_k = \frac{1}{d},$$

$$\text{if} \quad K > d \quad \text{and} \quad k \geq k_0, \qquad\qquad \eta_k = \frac{2}{(2d + k - k_0)},$$

*where $k_0 = \lceil \frac{K}{2} \rceil$.*

## B  IMPORTANT LEMMAS RELATED TO OUR ALGORITHMS

First we define the variables $\rho = (1 - \sqrt{1 - \lambda_2(W)})^K$ to characterize the effect of `FastMix`. Note that the setting of $K$ in Theorem 1 and 2 means

$$\rho^2 < \frac{1}{16}.$$

Then we introduce the following quantities:

- the global gradient estimation error: $U_t = \left\| \frac{1}{m} \sum_{i=1}^{m} (\mathbf{v}_{i,t} - \nabla f_i(\mathbf{x}_{i,t})) \right\|^2$;

- the local gradient estimation error: $V_t = \frac{1}{m} \|\mathbf{v}_t - \nabla \mathbf{f}(\mathbf{x}_t)\|^2$;

- the consensus error: $C_t = \|\mathbf{x_t} - 1\bar{x}_t\|^2$ and $Y_t = \frac{1}{m} \|\mathbf{y}_t - 1\bar{y}_t\|^2$.

At last we define two Lyapunov functions

$$\Phi_t = h_t + \frac{2\alpha}{(2 - p - 4\rho^2)L} Y_t + \frac{4\alpha}{pL} U_t + \frac{16\rho^2\alpha}{(2 - p - 4\rho^2)mL} V_t.$$

$$\Psi_t = h_t + \frac{\alpha}{(1 - 2\rho^2)L} Y_t + \frac{2\alpha}{pL} U_t + \frac{4\rho^2\alpha}{(1 - 2\rho^2)mL} V_t.$$

where $h_t$ is defined as $h_t := f(\bar{x}_t) - f(x^*)$. We describe the decrease of function value in following lemma.

**Lemma 8** *Suppose that each of $f_i$ and $x \in \mathcal{X}$ satisfy Assumption 1, 2, and 3. `DVRGTFW` holds that:*

$$h_{t+1} \leq (1 - \eta_t)h_t + \frac{\alpha}{L} Y_t + \frac{2\alpha}{L} U_t + \frac{2\alpha L}{m} C_t + \frac{\eta_t^2 L D^2 (\alpha + 2)}{2\alpha}. \tag{7}$$

*where $\alpha$ is some positive constant and $h_t$ is defined as $h_t := f(\bar{x}_t) - f(x^*)$.*

*Proof.* From the $L$-smoothness of global function $f$, we have:

$$f(\bar{x}_{t+1}) \leq f(\bar{x}_t) + \langle \nabla f(\bar{x}_t), \bar{x}_{t+1} - \bar{x}_t \rangle + \frac{L}{2} \|\bar{x}_{t+1} - \bar{x}_t\|^2,$$

where we use Lemma 6.

We subtract $f(x^*)$ in both sides, and by using the boundness of $\mathcal{C}$ we obtain the following inequality

$$h_{t+1} \le h_t + \frac{\eta_t}{m} \sum_{i=1}^m \langle \nabla f(\bar{x}_t), \mathbf{d}_{i,t} - \bar{x}_t \rangle + \frac{\eta_t^2 L D^2}{2}$$

$$\le h_t + \frac{\eta_t}{m} \sum_{i=1}^m \langle \mathbf{y}_{i,t}, \mathbf{d}_{i,t} - \bar{x}_t \rangle + \frac{\eta_t}{m} \sum_{i=1}^m \langle \nabla f(\bar{x}_t) - \mathbf{y}_{i,t}, \mathbf{d}_{i,t} - \bar{x}_t \rangle + \frac{\eta_t^2 L D^2}{2}$$

$$\le h_t + \frac{\eta_t}{m} \sum_{i=1}^m \langle \nabla f(\bar{x}_t), x^* - \bar{x}_t \rangle + \frac{\eta_t}{m} \sum_{i=1}^m \langle \nabla f(\bar{x}_t) - \mathbf{y}_{i,t}, \mathbf{d}_{i,t} - x^* \rangle + \frac{\eta_t^2 L D^2}{2}$$

$$\le h_t + \frac{\eta_t}{m} \sum_{i=1}^m \langle \nabla f(\bar{x}_t), x^* - \bar{x}_t \rangle + \frac{1}{m} \sum_{i=1}^m \langle \frac{\sqrt{\alpha}}{\sqrt{L}} (\nabla f(\bar{x}_t) - \mathbf{y}_{i,t}), \frac{\sqrt{L}\eta_t}{\sqrt{\alpha}} (\mathbf{d}_{i,t} - x^*) \rangle + \frac{\eta_t^2 L D^2}{2}$$

$$\le h_t + \frac{\eta_t}{m} \sum_{i=1}^m \langle \nabla f(\bar{x}_t), x^* - \bar{x}_t \rangle + \frac{\alpha}{2mL} \sum_{i=1}^m \|\nabla f(\bar{x}_t) - \mathbf{y}_{i,t}\|^2 + \frac{L\eta_t^2}{m\alpha} \sum_{i=1}^m \|\mathbf{d}_{i,t} - x^*\|^2 + \frac{\eta_t^2 L D^2}{2}$$

$$\le h_t - \eta_t(f(\bar{x}_t) - f(x^*)) + \frac{\alpha}{2mL} \sum_{i=1}^m (\nabla f(\bar{x}_t) - \mathbf{y}_{i,t})^2 + \frac{\eta_t^2 L D^2(\alpha+2)}{2\alpha}$$

$$\le (1 - \eta_t)h_t + \frac{\alpha}{L} \|\nabla f(\bar{x}_t) - \bar{v}_t\|^2 + \frac{\alpha}{mL} \|\mathbf{y}_t - \mathbf{1}\bar{y}_t\|^2 + \frac{\eta_t^2 L D^2(\alpha+2)}{2\alpha},$$

where we use the boundness of $\mathcal{X}$ and Lemma 3 in the first inequality; through the optimal choice of $\mathbf{d}_t$ which means that for each $i \in [m]$, $\langle \mathbf{y}_{i,t}, \mathbf{d}_{i,t} - \bar{x}_t \rangle \le \langle \mathbf{y}_{i,t}, x^* - \bar{x}_t \rangle$, and rearrange terms we get the third inequality; in the fifth inequality, we apply the Cauchy-Schwartz inequality to deduce $\langle \frac{\sqrt{a}}{\sqrt{L}} (\nabla f(\bar{x}_t) - \mathbf{y}_{i,t}), \frac{\sqrt{L}}{\sqrt{\alpha}} \eta_t (\mathbf{d}_{i,t} - x^*) \rangle \le \frac{\alpha}{2L} \|\nabla f(\bar{x}_t) - \mathbf{y}_{i,t}\|^2 + \frac{L\eta_t^2}{\alpha} \|\mathbf{d}_{i,t} - x^*\|^2$ with some positive constant $\alpha$; in the sixth inequality we use the boundness of $\mathcal{X}$ and Assumption 1; we apply Lemma 1 and Lemma 2 in the last inequality.

Now we consider decomposing the term $\|\nabla f(\bar{x}_t) - \bar{v}_t\|^2$. From the defination of $\nabla f(\bar{x}_t)$ and $\bar{v}_t$, the following inequality holds

$$\|\nabla f(\bar{x}_t) - \bar{v}_t\|^2$$

$$= \left\| \frac{1}{m} \sum_{i=1}^m (\nabla f_i(\bar{x}_t) - \mathbf{v}_{i,t}) \right\|^2$$

$$\le 2 \left\| \frac{1}{m} \sum_{i=1}^m (\nabla f_i(\bar{x}_t) - \nabla f_i(\mathbf{x}_{i,t})) \right\|^2 + 2 \left\| \frac{1}{m} \sum_{i=1}^m (\nabla f_i(\mathbf{x}_{i,t}) - \mathbf{v}_{i,t}) \right\|^2$$

$$\le 2 \left\| \frac{1}{m} \sum_{i=1}^m (\nabla f_i(\mathbf{x}_{i,t}) - \mathbf{v}_{i,t}) \right\|^2 + \frac{2}{m} \sum_{i=1}^m \|\nabla f_i(\mathbf{x}_{i,t}) - \nabla f_i(\bar{x}_t)\|^2$$

$$\le 2 \left\| \frac{1}{m} \sum_{i=1}^m (\nabla f_i(\mathbf{x}_{i,t}) - \mathbf{v}_{i,t}) \right\|^2 + \frac{2L^2}{m} \sum_{i=1}^m \|\mathbf{x}_{i,t} - \bar{x}_t\|^2$$

$$= 2 \left\| \frac{1}{m} \sum_{i=1}^m (\nabla f_i(\mathbf{x}_{i,t}) - \mathbf{v}_{i,t}) \right\|^2 + \frac{2L^2}{m} \|\mathbf{x}_t - \mathbf{1}\bar{x}_t\|^2$$

$$= 2U_t + \frac{2L^2}{m} C_t,$$

the first inequality uses Young's inequality; the second inequality uses Lemma 1; the third inequality based on the Assumption 2; in the last line we use the defination of $U_t$ and $C_t$. $\qquad \square$

Now we consider describe the decrease of $C_t, Y_t, U_t, V_t$ respectively. First, we provide the recursion for variable consensus error.

**Lemma 9** *Under the setting of Theorem 1, when $\rho^2 = \frac{1}{16}$, the following inequality holds*

$$C_t \leq \frac{8m\rho^2 D^2}{1 - 8\rho^2} \eta_t^2. \tag{8}$$

*Proof.* From the update of $\mathbf{x}_{t+1}$ in Algorithm 1, the following inequality holds

$$
\begin{aligned}
C_{t+1} &= \|\mathbf{x}_{t+1} - \mathbf{1}\bar{x}_{t+1}\|^2 \\
&= \left\| \text{FastMix}((1 - \eta_t)\mathbf{x}_t + \eta_t d_t, K_t) - \frac{1}{m}\mathbf{1}\mathbf{1}^\top \text{FastMix}(\mathbf{x}_t - \eta_t d_t, K_t) \right\|^2 \\
&\leq \rho^2 \left\| ((1 - \eta_t)\mathbf{x}_t - \eta_t \mathbf{d}_t) - \frac{1}{m}\mathbf{1}\mathbf{1}^\top ((1 - \eta_t)\mathbf{x}_t - \eta_t d_t) \right\|^2 \\
&= \rho^2 \left\| (1 - \eta_t)\mathbf{x}_t - \eta_t \mathbf{d}_t - (1 - \eta_t)\mathbf{1}(\bar{x}_t - \eta_t \bar{d}_t) \right\|^2 \\
&\leq 2\rho^2 (1 - \eta_t)^2 \|\mathbf{x}_t - \mathbf{1}\bar{x}_t\|^2 + 2\rho^2 \eta_t^2 \|\mathbf{d}_t - \mathbf{1}\bar{d}_t\|^2 \\
&\leq 2\rho^2 C_t + 2m\rho^2 \eta_t^2 D^2,
\end{aligned}
\tag{9}
$$

where we use Lemma 3 in the third inequality; in the fifith inequality we use Lemma 1; the last inequality based on the boundness of $\mathcal{X}$ and the defination of $C_t$.

From the setting of DVRGTFW, the following equality holds

$$C_0 = \|\mathbf{x}_0 - \mathbf{1}\bar{x}_0\|^2 = 0,$$

which satisfy Eq.(8), for the induction step, now we assume that $\forall t \geq 0$, Eq.(8) still holds, then we have the following inequality:

$$
\begin{aligned}
C_{t+1} &\leq 2\rho^2 C_t + 2m\rho^2 \eta_t^2 D^2 \\
&\leq \frac{2m\rho^2 D^2 \eta_t^2}{1 - 8\rho^2} \\
&\leq \frac{2m\rho^2 D^2 \eta_{t+1}^2}{1 - 8\rho^2} \frac{\eta_t^2}{\eta_{t+1}^2} \\
&\leq \frac{8m\rho^2 D^2 \eta_{t+1}^2}{1 - 8\rho^2},
\end{aligned}
$$

the last inequality is because from setting in Theorem 1, we can easily obtain $\max \frac{\eta_t^2}{\eta_{t+1}^2} \leq 4$, then we finish the proof. $\qquad \square$

Now we provide the recursion for gradient-tracking consensus error.

**Lemma 10** *Under the setting of Theorem 1, we have*

$$\mathbb{E}\left[Y_{t+1}\right] \leq 2\rho^2 \mathbb{E}\left[Y_t\right] + \frac{4}{m}\rho^2 p \mathbb{E}\left[V_t\right] + \frac{18}{m}\left(\frac{\rho^2(1-p)}{b} + 2\rho^2 p\right) L^2 \mathbb{E}\left[C_t\right] + 18\left(\frac{\rho^2(1-p)}{b} + 2\rho^2 p\right) L^2 \eta_t^2 D^2.$$

*Proof.* From the update of $\mathbf{v}_{i,t}$ in Algorithm 1, the following inequality holds

$$
\begin{aligned}
&\mathbb{E}\left[\|\mathbf{v}_{i,t+1} - \mathbf{v}_{i,t}\|^2\right] \\
&\leq p\mathbb{E}\|\nabla f_i(\mathbf{x}_{i,t+1}) - \mathbf{v}_{i,t}\|^2 + \frac{(1-p)}{b}\mathbb{E}\|\nabla f_{i,\xi_j}(\mathbf{x}_{i,t+1}) - \nabla f_{i,\xi_j}(\mathbf{x}_{i,t})\|^2 \\
&\leq 2p\mathbb{E}\|\nabla f_i(\mathbf{x}_{i,t+1}) - \nabla f_i(\mathbf{x}_{i,t})\|^2 + 2p\mathbb{E}\|\nabla f_i(\mathbf{x}_{i,t}) - \mathbf{v}_{i,t}\|^2 \\
&\quad + \frac{(1-p)L^2}{b}\mathbb{E}\|\mathbf{x}_{i,t+1} - \mathbf{x}_{i,t}\|^2 \\
&\leq 2pL^2 \mathbb{E}\|\mathbf{x}_{i,t+1} - \mathbf{x}_{i,t}\|^2 + 2p\mathbb{E}\|\nabla f_i(\mathbf{x}_{i,t}) - \mathbf{v}_{i,t}\|^2 \\
&\quad + \frac{(1-p)L^2}{b}\mathbb{E}\|\mathbf{x}_{i,t+1} - \mathbf{x}_{i,t}\|^2 \\
&= 2p\mathbb{E}\|\nabla f_i(\mathbf{x}_{i,t}) - \mathbf{v}_{i,t}\|^2 + \left(\frac{(1-p)}{b} + 2p\right) L^2 \mathbb{E}\|\mathbf{x}_{i,t+1} - \mathbf{x}_{i,t}\|^2,
\end{aligned}
\tag{10}
$$

the second inequality based on Young's inequality and the last inequality is due to Assumption 2.

Summing over Eq.(10) over $i \in [m]$, we obtain

$$
\begin{aligned}
\mathbb{E}&\left[\left\|\mathbf{v}_{t+1} - \mathbf{v}_t\right\|^2\right] \\
&\leq 2p\mathbb{E}\left\|\nabla \mathbf{f}(\mathbf{x}_t) - \mathbf{v}_t\right\|^2 + \left(\frac{(1-p)}{b} + 2p\right) L^2 \mathbb{E}\left\|\mathbf{x}_{t+1} - \mathbf{x}_t\right\|^2 \\
&\leq 2p\mathbb{E}\left\|\nabla \mathbf{f}(\mathbf{x}_t) - \mathbf{v}_t\right\|^2 \\
&\quad + 3\left(\frac{(1-p)}{b} + 2p\right) L^2 \mathbb{E}\left[\left\|\mathbf{x}_{t+1} - \mathbf{1}\bar{x}_{t+1}\right\|^2 + \left\|\mathbf{1}\bar{x}_{t+1} - \mathbf{1}\bar{x}_t\right\|^2 + \left\|\mathbf{x}_t - \mathbf{1}\bar{x}_t\right\|^2\right] \\
&\leq 2p\mathbb{E}\left\|\nabla \mathbf{f}(\mathbf{x}_t) - \mathbf{v}_t\right\|^2 \\
&\quad + 3\rho^2\left(\frac{(1-p)}{b} + 2p\right) L^2 (2\mathbb{E}\left[\left\|\mathbf{x}_t - \mathbf{1}\bar{x}_t\right\|^2\right] + 2m\eta_t^2 D^2) \\
&\quad + 3\left(\frac{(1-p)}{b} + 2p\right) L^2 \left(m\eta_t^2 D^2 + \mathbb{E}\left\|\mathbf{x}_t - \mathbf{1}\bar{x}_t\right\|^2\right) \\
&\leq 2pV_t + 9\left(\frac{(1-p)}{b} + 2p\right) L^2 C_t \\
&\quad + 9m\left(\frac{(1-p)}{b} + 2p\right) L^2 \eta_t^2 D^2,
\end{aligned}
\tag{11}
$$

where the second inequality based on Young's inequality; the third inequality uses the result of Eq.(9) and the boundness of $\mathcal{X}$.

From the update of $\mathbf{v}_t$ in DVRGTFW, we have

$$
\begin{aligned}
Y_{t+1} &= \frac{1}{m}\mathbb{E}\left[\left\|\mathbf{y}_{t+1} - \mathbf{1}\bar{y}_{t+1}\right\|^2\right] \\
&= \frac{1}{m}\mathbb{E}\left[\left\|\mathrm{FastMix}(\mathbf{y}_t + \mathbf{v}_{t+1} - \mathbf{v}_t, K) - \frac{1}{m}\mathbf{1}\mathbf{1}^\top \mathrm{FastMix}(\mathbf{y}_t + \mathbf{v}_{t+1} - \mathbf{v}_t, K)\right\|^2\right] \\
&\leq \frac{1}{m}\mathbb{E}\left[\rho^2\left\|\mathbf{y}_t + \mathbf{v}_{t+1} - \mathbf{v}_t - \frac{1}{m}\mathbf{1}\mathbf{1}^\top(\mathbf{y}_t + \mathbf{v}_{t+1} - \mathbf{v}_t)\right\|^2\right] \\
&\leq \frac{2}{m}\mathbb{E}\left[\rho^2\left\|\mathbf{y}_t - \mathbf{1}\bar{y}_t\right\|^2 + \rho^2\left\|\mathbf{v}_{t+1} - \mathbf{v}_t - \frac{1}{m}\mathbf{1}\mathbf{1}^\top(\mathbf{v}_{t+1} - \mathbf{v}_t)\right\|^2\right] \\
&\leq \frac{2}{m}\mathbb{E}\left[\rho^2\left\|\mathbf{y}_t - \mathbf{1}\bar{y}_t\right\|^2\right] + \frac{2}{m}\mathbb{E}\left[\rho^2\left\|\mathbf{v}_{t+1} - \mathbf{v}_t\right\|^2\right] \\
&\leq 2\rho^2 Y_t + \frac{4}{m}\rho^2 p V_t + \frac{18}{m}\left(\frac{\rho^2(1-p)}{b} + 2\rho^2 p\right) L^2 C_t \\
&\quad + 18\left(\frac{\rho^2(1-p)}{b} + 2\rho^2 p\right) L^2 \eta_t^2 D^2,
\end{aligned}
\tag{12}
$$

where we use Lemma 3 in the first inequality; in the second inequality we use Young's inequality; we apply Lemma 4 in the third inequality; combine the result in Eq.(11) and the defination of $Y_t$, then we finish the proof. $\qquad\square$

Now we provide the recursion for local and global error of gradient estimation.

**Lemma 11** *Under the setting of Theorem 1, we have*

$$
V_{t+1} \leq (1-p)\mathbb{E}\left[V_t + \frac{3L^2(1+2\rho^2)}{mb}C_t + \frac{3L^2(1+2\rho^2)\eta_t^2 D^2}{b}\right].
$$

*Proof.* The update of $\mathbf{v}_{i,t}$ means

$$\mathbb{E}\left\|\mathbf{v}_{i,t} - \nabla f_i(\mathbf{x}_{i,t+1})\right\|^2$$

$$= p\mathbb{E}\left\|\nabla f_i(\mathbf{x}_{i,t+1}) - \nabla f_i(\mathbf{x}_{i,t+1})\right\|^2$$

$$+ (1-p)\mathbb{E}\left\|\mathbf{v}_{i,t} + \frac{1}{b}\sum_{j=1}^{b}\left(\nabla f_{i,\xi_j}(\mathbf{x}_{i,t+1}) - \nabla f_{i,\xi_j}(\mathbf{x}_{i,t})\right) - \nabla f_i(\mathbf{x}_{i,t+1})\right\|^2$$

$$= (1-p)\mathbb{E}\left\|\mathbf{v}_{i,t} - \nabla f_i(\mathbf{x}_{i,t})\right\|^2 \tag{13}$$

$$+ (1-p)\mathbb{E}\left\|\frac{1}{b}\sum_{j=1}^{b}\left(\nabla f_{i,\xi_j}(\mathbf{x}_{i,t+1}) - \nabla f_{i,\xi_j}(\mathbf{x}_{i,t})\right) - \nabla f_i(\mathbf{x}_{i,t+1}) + \nabla f_i(\mathbf{x}_{i,t})\right\|^2$$

$$\leq (1-p)\mathbb{E}\left\|\mathbf{v}_{i,t} - \nabla f_i(\mathbf{x}_{i,t})\right\|^2 + \frac{1-p}{b}\mathbb{E}\left\|\nabla f_{i,\xi_j}(\mathbf{x}_{i,t+1}) - \nabla f_{i,\xi_j}(\mathbf{x}_{i,t})\right\|^2$$

$$\leq (1-p)\mathbb{E}\left\|\mathbf{v}_{i,t} - \nabla f_i(\mathbf{x}_{i,t})\right\|^2 + \frac{(1-p)L^2}{b}\mathbb{E}\left\|\mathbf{x}_{i,t+1} - \mathbf{x}_{i,t}\right\|^2,$$

where the first inequality based on the update of $\mathbf{v}_{i,t}$ in DVRGTFW; the second equality uses the property of Martingale (Proposition 1 from Fang et al. (2018)); the first inequality use the property of variance and independence of $\xi_1, \ldots, \xi_b$; the last step based on Assumption 2.

Taking the average over on above result over $i = 1, \ldots, m$, we obtain

$$\mathbb{E}[V_{t+1}] = \frac{1}{m}\mathbb{E}\left\|\mathbf{v}_{t+1} - \nabla\mathbf{f}(\mathbf{x}_{t+1})\right\|^2$$

$$\leq \frac{1-p}{m}\mathbb{E}\left[\left\|\mathbf{v}_t - \nabla\mathbf{f}(\mathbf{x}_t)\right\|^2\right] + \frac{(1-p)L^2}{mb}\mathbb{E}\left[\left\|\mathbf{x}_{t+1} - \mathbf{x}_t\right\|^2\right]$$

$$\leq \frac{1-p}{m}\mathbb{E}\left\|\mathbf{v}_t - \nabla\mathbf{f}(\mathbf{x}_t)\right\|^2 \tag{14}$$

$$+ \frac{3(1-p)L^2}{mb}\mathbb{E}\left[\left\|\mathbf{x}_{t+1} - \mathbf{1}\bar{x}_{t+1}\right\|^2 + \mathbb{E}\left\|\mathbf{1}\bar{x}_{t+1} - \mathbf{1}\bar{x}_t\right\|^2 + \mathbb{E}\left\|\mathbf{x}_t - \mathbf{1}\bar{x}_t\right\|^2\right]$$

$$\leq (1-p)\mathbb{E}\left[V_t + \frac{3L^2(1+2\rho^2)}{mb}C_t + \frac{3L^2(1+2\rho^2)\eta_t^2 D^2}{b}\right].$$

$\square$

**Lemma 12** *Under the setting of Theorem 1, we have*

$$\mathbb{E}[U_{t+1}] \leq (1-p)\mathbb{E}\left[U_t + \frac{3L^2(1+2\rho^2)}{m^2 b}C_t + \frac{3(1+2\rho^2)L^2\eta_t^2 D^2}{mb}\right].$$

*Proof.* The update of $\mathbf{v}_{i,t}$ means

$$\mathbb{E}[U_{t+1}]$$

$$= p\mathbb{E}\left\|\frac{1}{m}\sum_{i=1}^{m}\left(\nabla f_i(\mathbf{x}_{i,t+1}) - \nabla f_i(\mathbf{x}_{i,t+1})\right)\right\|^2$$

$$+ (1-p)\mathbb{E}\left\|\frac{1}{m}\sum_{i=1}^{m}\left(\mathbf{v}_{i,t} + \frac{1}{b}\sum_{j=1}^{b}\left(\nabla f_{i,\xi_j}(\mathbf{x}_{i,t+1}) - \nabla f_{i,\xi_j}(\mathbf{x}_{i,t})\right) - \nabla f_i(\mathbf{x}_{i,t+1})\right)\right\|^2 \tag{15}$$

$$= (1-p)\mathbb{E}\left\|\frac{1}{m}\sum_{i=1}^{m}\left(\mathbf{v}_{i,t} - \nabla f_i(\mathbf{x}_{i,t})\right)\right\|^2$$

$$+ (1-p)\mathbb{E}\left\|\frac{1}{mb}\sum_{i=1}^{m}\sum_{j=1}^{b}\left(\left(\nabla f_{i,\xi_j}(\mathbf{x}_{i,t+1}) - \nabla f_{i,\xi_j}(\mathbf{x}_{i,t})\right) - \nabla f_i(\mathbf{x}_{i,t+1}) + \nabla f_i(\mathbf{x}_{i,t})\right)\right\|^2$$

$$\leq (1-p)\mathbb{E}[U_t] + \frac{1-p}{m^2b^2} \sum_{i=1}^m \sum_{j=1}^b \mathbb{E} \left\| \nabla f_{i,\xi_j}(\mathbf{x}_{i,t+1}) - \nabla f_{i,\xi_j}(\mathbf{x}_{i,t}) \right\|^2$$

$$\leq (1-p)\mathbb{E}[U_t] + \frac{(1-p)L^2}{m^2b^2} \sum_{i=1}^m \sum_{j=1}^b \mathbb{E} \left\| \mathbf{x}_{i,t+1} - \mathbf{x}_{i,t} \right\|^2$$

$$= (1-p)\mathbb{E}[U_t] + \frac{(1-p)L^2}{m^2b} \mathbb{E} \left\| \mathbf{x}_{t+1} - \mathbf{x}_t \right\|^2 \tag{16}$$

$$\leq (1-p)\mathbb{E}[U_t] + \frac{3(1-p)L^2}{m^2b} \mathbb{E}[\|\mathbf{x}_{t+1} - \mathbf{1}x_{t+1}\|^2 + \|\mathbf{1}x_{t+1} - \mathbf{1}x_t\|^2 + \|\mathbf{x}_t - \mathbf{1}x_t\|^2]$$

$$\leq (1-p)\mathbb{E}\left[ U_t + \frac{6\rho^2 L^2}{m^2b}C_t + \frac{6\rho^2 L^2 \eta_t^2 D^2}{mb} + \frac{3L^2}{mb}\|\bar{x}_{t+1} - \bar{x}_t\|^2 + \frac{3L^2}{m^2b}C_t \right]$$

$$\leq (1-p)\mathbb{E}\left[ U_t + \frac{3L^2(1+2\rho^2)}{m^2b}C_t + \frac{3(1+2\rho^2)L^2\eta_t^2 D^2}{mb} \right],$$

where the second equality use the property of Martingale; the first inequality based on the property of variance and independence of $\xi_1, \ldots, \xi_b$; the second inequality based on Assumption 2; the third inequality use Young's inequality; the fourth inequality use Eq.(9), the last two steps use the boundness of $\mathcal{X}$ and the defination of $C_t$, then we finish the proof. $\qquad\square$

## C  PROOF OF THEOREM 1

*Proof.* From the defination of $\phi_t$ and combing results of Lemma 8, 10, 11 and 12, we have

$$\mathbb{E}\left[\Phi_{t+1}\right]$$

$$= \mathbb{E}\left[ h_{t+1} + \frac{2\alpha}{(2-p-4\rho^2)L}Y_{t+1} + \frac{4\alpha}{pL}U_{t+1} + \frac{16\rho^2\alpha}{(2-p-4\rho^2)mL}V_{t+1} \right]$$

$$\leq \mathbb{E}\left[ (1-\eta_t)h_t + (1-\frac{p}{2})\frac{2\alpha}{(2-p-4\rho^2)L}Y_t + (1-\frac{p}{2})\frac{4\alpha}{pL}U_t + (1-\frac{p}{2})\frac{16\rho^2\alpha}{(2-p-4\rho^2)mL}V_t \right]$$

$$+ \left(\frac{2\alpha L}{m} + \frac{36\alpha L}{(2-p-4\rho^2)n}\left(\frac{\rho^2(1-p)}{b} + 2\rho^2 p\right) + \frac{3(1-p)L(1+2\rho^2)(8\alpha - 4p\alpha - 16\rho^2\alpha + 16\rho^2 p\alpha)}{(2-p-4\rho^2)m^2bp}\right)\mathbb{E}[C_t]$$

$$+ \left(\frac{L(\alpha+2)}{2\alpha} + \frac{36\alpha L}{2-p-4\rho^2}\left(\frac{\rho^2(1-p)}{b} + 2\rho^2 p\right) + \frac{3(1-p)L(1+2\rho^2)(8\alpha - 4p\alpha - 16\rho^2\alpha + 16\rho^2 p\alpha)}{(2-p-4\rho^2)mbp}\right)D^2\eta_t^2$$

$$\leq \max\{1-\eta_t, 1-\frac{p}{2}\}\mathbb{E}\left[ h_t + \frac{2\alpha}{(7-4p)L}Y_t + \frac{1}{pL}U_t + \frac{\alpha}{(7-4p)mL}V_t \right] + 16\alpha LD^2\eta_t^2 + \frac{LD^2\eta_t^2}{\alpha}.$$

The second inequality based on Lemma 9 and the settings of $p$, $b$ and $K$ in Theorem 1. If we choose $\eta_t \leq \frac{p}{2}$ and $\alpha = \frac{1}{4}$, then we will have

$$\mathbb{E}[\Phi_{t+1}] \leq (1-\eta_t)\mathbb{E}[\Phi_t] + 8LD^2\eta_t^2.$$

It remains to use Lemma 7 with $c = 8LD^2$, $d = \frac{2}{p}$ and then we obtain

$$\mathbb{E}[f(\bar{x}_t) - f(x^*) + \frac{1}{2(7-4p)mL}Y_t + \frac{1}{pL}U_t + \frac{1}{(7-4p)mL}V_t]$$

$$= \mathcal{O}\left( \frac{1}{p}\left( f(\bar{x}_0) - f(x^*) + \frac{1}{2(7-4p)mL}\mathbb{E}\left[Y_0\right] \right) \exp\left( -\frac{pT}{4} \right) + \frac{8LD^2}{T} \right).$$

From DVRGTFW, it is easy to obtain $\mathbb{E}\left[V_0\right] = 0$ and $\mathbb{E}\left[U_0\right] = 0$. Using the setting of $K_{in}$ in Theorem 1, we can deduce

$$\frac{1}{2(7-4p)L}\mathbb{E}\left[Y_0\right] \leq \frac{1}{2(7-4p)L}(1 - \sqrt{1-\lambda_2(W)})^{2K_{in}}\|\mathbf{v}^0 - \mathbf{1}\bar{v}^0\|^2$$

$$\leq \frac{1}{2(7-4p)} \leq \frac{1}{6}.$$

Then we obtain

$$\mathbb{E}[f(\bar{x}_t) - f(x^*)] = \mathcal{O}\left(\frac{1}{p}\left(f(\bar{x}_0) - f(x^*) + \frac{1}{6}\right)\exp\left(-\frac{pT}{4}\right) + \frac{8LD^2}{T}\right).$$

$\square$

## D  PROOF OF THEOREM 2

*Proof.* According to Assumption 2 and Lemma 6, we have

$$f(\bar{x}_{t+1}) \leq f(\bar{x}_t) + \langle \nabla f(\bar{x}_t), \bar{x}_{t+1} - \bar{x}_t \rangle + \frac{L}{2}\|\bar{x}_{t+1} - \bar{x}_t\|^2.$$

Subtracting $f(x^*)$ from both sides, we get

$$h_{t+1} \leq h_t + \frac{\eta_t}{m}\sum_{i=1}^{m}\langle \nabla f(\bar{x}_t), \mathbf{d}_{i,t} - \bar{x}_t \rangle + \frac{\eta_t^2 LD^2}{2}$$

$$\leq h_t + \frac{\eta_t}{m}\sum_{i=1}^{m}\langle \mathbf{y}_{i,t}, \mathbf{d}_{i,t} - \bar{x}_t \rangle + \frac{\eta_t}{m}\sum_{i=1}^{m}\langle \nabla f(\bar{x}_t) - \mathbf{y}_{i,t}, \mathbf{d}_{i,t} - \bar{x}_t \rangle + \frac{\eta_t^2 LD^2}{2}$$

$$\leq h_t + \frac{\eta_t}{m}\sum_{i=1}^{m}\langle \nabla f(\bar{x}_t), x - \bar{x}_t \rangle + \frac{\eta_t}{m}\sum_{i=1}^{m}\langle \nabla f(\bar{x}_t) - \mathbf{y}_{i,t}, \mathbf{d}_{i,t} - x^* \rangle + \frac{\eta_t^2 LD^2}{2}$$

$$\leq h_t + \frac{\eta_t}{m}\sum_{i=1}^{m}\langle \nabla f(\bar{x}_t), x - \bar{x}_t \rangle + \frac{1}{m}\sum_{i=1}^{m}\langle \frac{\sqrt{\alpha}}{\sqrt{L}}(\nabla f(\bar{x}_t) - \mathbf{y}_{i,t}), \frac{\sqrt{L}\eta_t}{\sqrt{\alpha}}(\mathbf{d}_{i,t} - x^*) \rangle + \frac{\eta_t^2 LD^2}{2}$$

$$\leq h_t + \frac{\eta_t}{m}\sum_{i=1}^{m}\langle \nabla f(\bar{x}_t), x - \bar{x}_t \rangle + \frac{\alpha}{2L}\sum_{i=1}^{m}\|\nabla f(\bar{x}_t) - \mathbf{y}_{i,t}\|^2 + \frac{L\eta_t^2}{m\alpha}\sum_{i=1}^{m}\|\mathbf{d}_{i,t} - x^*\|^2 + \frac{\eta_t^2 LD^2}{2}$$

$$\leq h_t + \eta_t\langle \nabla f(\bar{x}_t), x - \bar{x}_t \rangle + \frac{\alpha}{L}\|\nabla f(\bar{x}_t) - \bar{v}_t\|^2 + \frac{\alpha}{mL}\|\mathbf{y}_t - \mathbf{1}\bar{y}_t\|^2 + \frac{\eta_t^2 LD^2(\alpha+2)}{2\alpha}$$

$$\leq h_t + \eta_t\langle \nabla f(\bar{x}_t), x - \bar{x}_t \rangle + \frac{\alpha}{L}Y_t + \frac{2\alpha}{L}U_t + \frac{2\alpha L}{m}C_t + \frac{\eta_t^2 LD^2(\alpha+2)}{2\alpha}. \tag{17}$$

We omit the explanation of the proof because it's similar to the proof of Lemma 8, then we rearrange the term in Eq.(17), we obtain

$$\eta_t\langle \nabla f(\bar{x}_t), \bar{x}_t - x \rangle \leq h_t - h_{t+1} + \frac{\alpha}{L}Y_t + \frac{2\alpha}{L}U_t + \frac{2\alpha L}{m}C_t + \frac{\eta_t^2 LD^2(\alpha+2)}{2\alpha}.$$

Maximizing over all $x \in \mathcal{X}$ and take the full mathematical expectation, we get

$$\mathbb{E}\left[\max_{x\in\mathcal{X}}\langle \nabla f(\bar{x}_t), \bar{x}_t - x \rangle\right] \leq \mathbb{E}\left[\psi_t - \psi_{t+1}\right]$$

$$+ \left(\frac{2\alpha L}{m} + \frac{18\alpha L}{(1-2\rho^2)m}\left(\frac{\rho^2(1-p)}{b} + 2\rho^2 p\right) + \frac{3(1-p)L(1+2\rho^2)(4\rho^2\alpha p - 4\rho^2\alpha + 2\alpha)}{m^2 bp(1-2\rho^2)}\right)\mathbb{E}\left[C_t\right]$$

$$+ \left(\frac{L(\alpha+2)}{2\alpha} + \frac{18\alpha L}{1-2\rho^2}\left(\frac{\rho^2(1-p)}{b} + 2\rho^2 p\right) + \frac{3(1-p)L(1+2\rho^2)(4\rho^2\alpha p - 4\rho^2\alpha + 2\alpha)}{mbp(1-2\rho^2)}\right)D^2\eta_t^2.$$

$$\leq \mathbb{E}[\phi_t - \phi_{t+1}] + 8\alpha LD^2\eta_t^2 + \frac{LD^2\eta_t^2}{\alpha} + \frac{L}{2} \leq \mathbb{E}[\phi_t - \phi_{t+1}] + 7LD^2\eta_t^2.$$

In the first inequality we use the defination of $\psi_t$; the second inequality based on Lemma 9 and the settings of $p$, $b$ and $K$ in the Theorem 2; We obtain the last inequality with the choice of $\alpha = \frac{1}{2\sqrt{2}}$.

Summing over all $t$ from 0 to $T-1$, we have

$$\sum_{t=0}^{T-1} \eta_t \mathbb{E}\left[\max_{x\in\mathcal{X}}\langle\nabla f(\bar{x}_t), \bar{x}_t - x\rangle\right] \leq \mathbb{E}\left[\psi_0 - \psi_T\right] + 7LD^2 \sum_{t=0}^{T-1}\eta_t^2$$

$$\leq \mathbb{E}\left[\psi_0\right] + 7LD^2 \sum_{t=0}^{T-1}\eta_t^2$$

$$\leq \mathbb{E}\left[f(\bar{x}_0) - f(x^*) + \frac{2\sqrt{2}}{7}\right] + 7LD^2 \sum_{t=0}^{T-1}\eta_t^2.$$

The last inequality based on the setting of $K_{in}$ in Theorem 2. If we take $\eta_t = \frac{1}{\sqrt{T}}$ and devide both sides by $\sqrt{T}$, then

$$\mathbb{E}\left[\frac{1}{T}\sum_{t=0}^{T-1}\max_{x\in\mathcal{X}}\langle\nabla f(\bar{x}_t), \bar{x}_t - x\rangle\right] \leq \frac{f(\bar{x}_0) - f(x^*) + \frac{2\sqrt{2}}{7}}{\sqrt{T}} + \frac{7LD^2}{\sqrt{T}}.$$

$\square$

