# OpenReview forum: "A Computation and Communication Efficient Projection-free Algorithm for Decentralized Constrained Optimization"
_ICLR.cc/2025/Conference — ICLR 2025 Conference Withdrawn Submission_

### Official Review · Reviewer_yJBK · 2024-10-15

**Soundness:** 3
**Presentation:** 2
**Contribution:** 2
**Rating:** 5
**Confidence:** 3

**Summary:**

In this paper, the authors proposed to combine the Frank-Wolfe algorithm with variance reduction as well as gradient tracking in the decentralized setting, resulting in the algorithm DVRGTFW. Convergence analysis in the convex and non-convex case are provided with numerical experiments conducted to further support the theory provided.

**Strengths:**

1. The author manages to combine the technique of variance reduction and gradient tracking to Frank-Wolfe algorithm in the decentralized setting, convergence analysis in both convex case and non-convex case are provided, illustrating the effectiveness of the proposed algorithm DVRGTFW.

2. The proposed algorithm achieves best-known incremental first order oracle complexities both in the convex case and in the non-convex case, and near optimal communication complexity in the non-convex case.

3. The paper offers numerical experiments to validate the theory presented in the paper.

**Weaknesses:**

1. Though the results are interesting, the proposed method appears to be primarily a combination of established techniques, such as variance reduction, gradient tracking, and the Frank-Wolfe algorithm. As a result, the novelty of the approach may be somewhat limited.

2. If I am not mistaken, the communication complexity for DVRGTFW is not better than existing methods in the convex case given its extra dependence on $\sqrt{mn}$ as it is demonstrated in Table 1, which is a limitation of the algorithm.

3. I recommend that the authors do a thorough check of the paper as there are many typos, some of them are confusing, such examples include:
- At line 92, ''develop communication and communication efficient'';
- At line 114, $m = 0$;
- At line 222, $x_0 \in \mathbb{R}^d$,
- There are also some notations used without introduction in the paper.

4. In some of the numerical experiments, the proposed algorithm is not better than existing algorithm for an unclear reason.

**Questions:**

1. In table 1, when $m = 1$, we should recover the complexities in the centralized setting in the convex/non-convex setting, however, for the proposed algorithm, the reviewer does not understand why it matches the bounds given in [Beznosikov et al., 2023], for example, in the convex case the table suggests $\tilde{\mathcal{O}}(n + \frac{\sqrt{n}}{\varepsilon})$, while [Beznosikov et al., 2023] gives $\tilde{\mathcal{O}}(n + \frac{1}{\varepsilon})$.

2. What is the output of Algorithm 2 FastMix?

3. Is it possible to further improve the communication complexity of the algorithm so that it matches the optimal bounds?

---

### Official Review · Reviewer_jSAK · 2024-10-31

**Soundness:** 2
**Presentation:** 2
**Contribution:** 2
**Rating:** 3
**Confidence:** 5

**Summary:**

This paper develops a decentralized stochastic Frank-Wolfe algorithm and establishes its convergence rate for both convex and nonconvex constrained problems. The experiment demonstrates the effectiveness of the proposed algorithm.

**Strengths:**

1. This paper is well written. It is easy to follow.

2. The literature review is good.

**Weaknesses:**

1. The novelty is limited. Decentralized unconstrained optimization has been well studied. This paper tries to extend those algorithms to constrained problem, where the feasible set is bounded. However, this extension is trivial. In particular, due to the bounded feasible set, it is trivial to bound the gradient variance. Actually, the proof for frank-wolfe algorithm is much easier than the unconstrained counterpart.

2. As mentioned in this paper, there are some existing decentralized Frank-wolfe algorithms for DR-submodular optimization problems. What is the difference between those algorithms and this paper? Are there any unique challenges compared to those algorithms? It would be good if the authors could discuss these critical points to show the contribution of this paper.

3. FastMix is a not very common communication method. It would be good to provide some background for this method. For example, in standard gradient tracking method, it is well known that $\bar{v}_t=\bar{y}_t$. Does FastMix also have this property? It seems the authors directly use $\bar{v}_t=\bar{y}_t$ in the proof.

4. It would be good to provide more details about the proof. For example, how to get the third step in Line 764? It is not very clear.

5. How does the heterogeneity affect the convergence rate?

6. Why does IFO not depend on the spectral gap? Any explanation?

**Questions:**

Please see Weakness.

---

> ### Author Response · Authors · 2024-11-14
>
> 1.Decentralized constrained optimization is definitely a non-trivial problem. Starting from the initial distributed unconstrained methods with linear convergence rates [7], it took approximately six years to propose the first distributed gradient tracking proximal algorithm with linear convergence rates [8]. However, as mentioned in our paper, projection into complex sets, such as trace norms , can be computationally expensive. For example, for the trace norm constraint, FW can leverage the power method to efficiently obtain only the largest eigenvalue of the matrix, avoiding the costly computation of a full SVD [see this note, page10] (https://www.stat.cmu.edu/~ryantibs/convexopt-F18/lectures/frank-wolfe.pdf). To address the issue of high computational costs, methods such as the Frank-Wolfe algorithm can be utilized to find solutions in the constraint set via a linear oracle, which is usually cheaper in computation cost than using a direct projection.
>
> Regarding the bounded set assumption you mentioned, it is a standard assumption in Frank-Wolfe type algorithms, which applied in both centralized setting [9] and distributed setting [10]. Additionlly, consindering Question 5, we think that the mentioned term ``gradient variance`` refers to $\sum_{i=1}^m \Vert \nabla f_i(x) - \nabla f(x)\Vert^2$, which measures the variance between nodes' gradients and the global one. While, this gradient variance **does not** need to considered in our method due to the use of gradient tracking technique, which utilizes bias correction to compensate heterogeneous gradient [18] and can overcome the heterogeneity. Hence, our proof **does not** rely on this assumption. If this gradient variance is considered as $\sum_{i=1}^m \Vert \nabla f_{i, \xi}(x) - \nabla f_i(x)\Vert^2$, which arises from stochasity, this is a stand assumption in stochastic optimization, regardless of whether the problem is constrained or unconstrained. Therefore, we do not see any reason that this bounded gradient variance render our proof trivial.
>
> Besides, the proof of Frank-wolfe is not easier than its counterpart, e.g., stochastic gradient descent (SGD).
> For instance, as shown in [19, 20], the stochastic frank-wolfe is actually more complexty than SGD due to the need for an additional gradient estimator to ensure convergence. The analysis of the bound between this gradient estimator and the true gradient is more intricate compared to that in SGD.

---

> > ### Author Response · Authors · 2024-11-14
> >
> > 2.The DR-submodular problem is indeed mentioned in the article, but we only use it as an example in the related work section to  demonstrate the development of the decentralized Frank-Wolfe algorithm for this problem. In fact, the DR-submodular problem is just a subset of our broader problem. The DR-submodular problem makes additional assumptions about the function, which our paper does not require. We are comparing more general algorithms, such as DeFW [10], DstoFW [11], DMFW [12] and I-PDS [13] (the assumption on the constraint sets has some differences). The contribution of this article has been highlighted in the contribution section of the introduction, this article has the best-known IFO complexity both in the convex case and the non-convex case and nearly optimal communication complexity in non-convex case compared to the previous decentralized stochastic Frank-Wolfe algorithm. The contribution is significant in variance-reduction type problems.
> >
> > 3.Acceleration method is commonly used in distributed algorithms to reduce the dependence on spectral gap for the convergence speed, for more details, please refer to [14] , if you need it, we can add a section about acceleration method in the introduction. Regarding $\bar{y}_t=\bar{v}_t$, we have mentioned it in Lemma 2 in Appendix A.
> >
> > 4.In line 778-779, we have briefly outlined the derivation of the proof in lines 764. The proof process goes as follows: starting from the update of $\mathbf{d}\_t=argmin\_{\mathbf{d}\in\mathcal{X}}\langle \mathbf{y}\_{t},d\rangle$ in Algorithm 1, we can easily deduce that for each $i\in[m]$, $\langle \mathbf{y}\_{i,t},\mathbf{d}\_{i,t}-\bar{x}\_t \rangle \leq \langle \mathbf{y}\_{i,t},x^*-\bar{x}\_t \rangle$, then we use the term $\langle \mathbf{y}\_{i,t},x^*-\bar{x}\_t \rangle$ to substitute the term $\langle \mathbf{y}\_{i,t},\mathbf{d}\_{i,t}-\bar{x}\_t \rangle$, add it to the third term $\langle \nabla f(\bar{x}\_{t})-\mathbf{y}\_{i,t},\mathbf{d}\_{i,t}-\bar{x}\_{t}\rangle$ in line 762 and sum up to obtain the result in line 764.
> >
> > 5.A key feature of gradient tracking is a tracking mechanism that allows to overcome data heterogeneity between nodes [18].
> >
> > 6.Please refer to the third point, the acceleration algorithm [15,16,17] can reduce the dependence on spectral gap for the convergence speed, and as we have mentioned in Remark 1, the multi-consensus step in FastMix enable our analysis closed to the centralized algorithm.  Moreover, the acceleration algorithm was not initially introduced in this work, we refrain from providing extensive insights into its mechanics.
> >
> > References
> >
> > [1] Ling, Qing, et al. "Decentralized low-rank matrix completion." 2012 IEEE International Conference on Acoustics, Speech and Signal Processing (ICASSP). IEEE, 2012.
> >
> > [2] Mackey, Lester W., Ameet Talwalkar, and Michael I. Jordan. "Distributed matrix completion and robust factorization." J. Mach. Learn. Res. 16.1 (2015): 913-960.
> >
> > [3] Yu, Hsiang-Fu, et al. "Scalable coordinate descent approaches to parallel matrix factorization for recommender systems." 2012 IEEE 12th international conference on data mining. IEEE, 2012.
> >
> > [4] Lacoste-Julien, Simon. "Convergence rate of frank-wolfe for non-convex objectives." arXiv preprint arXiv:1607.00345 (2016).
> >
> > [5] Lafond, Jean, Hoi-To Wai, and Eric Moulines. "On the online Frank-Wolfe algorithms for convex and non-convex optimizations." arXiv preprint arXiv:1510.01171 (2015).
> >
> > [6] Duchi, John, et al. "E cient projections onto the 1-ball for learning in high dimensions." Proceedings of the 25th International.
> >
> > [7] Shi, Wei, et al. "Extra: An exact first-order algorithm for decentralized consensus optimization." SIAM Journal on Optimization 25.2 (2015): 944-966.
> >
> > [8] Alghunaim, Sulaiman, Kun Yuan, and Ali H. Sayed. "A linearly convergent proximal gradient algorithm for decentralized optimization." Advances in Neural Information Processing Systems 32 (2019).
> >
> > [9] Jaggi, Martin. "Revisiting Frank-Wolfe: Projection-free sparse convex optimization." International conference on machine learning. PMLR, 2013.
> >
> > [10] Wai, Hoi-To, et al. "Decentralized Frank–Wolfe algorithm for convex and nonconvex problems." IEEE Transactions on Automatic Control 62.11 (2017): 5522-5537.
> >
> > [11] Jiang, Xia, et al. "Distributed stochastic projection-free solver for constrained optimization." arXiv preprint arXiv:2204.10605 (2022).
> >
> > [12] Hou, Jie, et al. "Distributed momentum-based Frank-Wolfe algorithm for stochastic optimization." IEEE/CAA Journal of Automatica Sinica 10.3 (2022): 685-699.
> >
> > [13] Nguyen, Hoang Huy, Yan Li, and Tuo Zhao. "Stochastic Constrained Decentralized Optimization for Machine Learning with Fewer Data Oracles: a Gradient Sliding Approach." arXiv preprint arXiv:2404.02511 (2024).
> >
> > [14] d’Aspremont, Alexandre, Damien Scieur, and Adrien Taylor. "Acceleration methods." Foundations and Trends® in Optimization 5.1-2 (2021): 1-245.

---

> > > ### Author Response · Authors · 2024-11-14
> > >
> > > [15] Qu, Guannan, and Na Li. "Accelerated distributed Nesterov gradient descent." IEEE Transactions on Automatic Control 65.6 (2019): 2566-2581.
> > >
> > > [16] Li, Huan, and Zhouchen Lin. "Revisiting extra for smooth distributed optimization." SIAM Journal on Optimization 30.3 (2020): 1795-1821.
> > >
> > > [17] Ye, Haishan, et al. "Multi-consensus decentralized accelerated gradient descent." Journal of Machine Learning Research 24.306 (2023): 1-50.
> > >
> > > [18] Liu, Yue, et al. "Decentralized gradient tracking with local steps." Optimization Methods and Software (2024): 1-28.
> > >
> > > [19] Mokhtari, Aryan, Hamed Hassani, and Amin Karbasi. "Conditional gradient method for stochastic submodular maximization: Closing the gap." International Conference on Artificial Intelligence and Statistics. PMLR, 2018.
> > >
> > > [20] Sahu, Anit Kumar, Manzil Zaheer, and Soummya Kar. "Towards gradient free and projection free stochastic optimization." The 22nd International Conference on Artificial Intelligence and Statistics. PMLR, 2019.

---

> > > > ### Author Response · Authors · 2024-11-23
> > > >
> > > > If we have addressed your concerns, please consider raising the score, as the deadline is approaching.

---

> > > > > ### Comment · Reviewer_jSAK · 2024-11-27
> > > > >
> > > > > Thank for addressing most of my concerns. However, the major issue, limited novelty, still remains. I will keep my score.

---

> > > > > > ### Author Response · Authors · 2024-11-29
> > > > > >
> > > > > > Dear Reviewer jSAK,
> > > > > >
> > > > > > Thank you for your response.
> > > > > >
> > > > > > Could you please clarify your concerns regarding the novelty? The term "novelty" encompasses a broad range of meanings, and we would appreciate it if you could specify the particular aspects.
> > > > > >
> > > > > > In our previous response, we have claimed that the decentralized constrained optimization problem is not a trivial extension of its unconstrained counterpart. The assumption of a bounded feasible set is fundenmental and common for the Frank-Wolfe method.
> > > > > > Moreover, prior studies [1, 2] have provided overly complex and cumbersome convergence analyses, yielding suboptimal bounds.
> > > > > >
> > > > > > If you could further clarify what specific aspects of "novelty" you are referring to, we will provide detailed explanations.
> > > > > >
> > > > > > References
> > > > > >
> > > > > > [1] Wai, Hoi-To, et al. "Decentralized Frank–Wolfe algorithm for convex and nonconvex problems." IEEE Transactions on Automatic Control 62.11 (2017): 5522-5537.
> > > > > >
> > > > > > [2] Hou, Jie, et al. "Distributed momentum-based Frank-Wolfe algorithm for stochastic optimization." IEEE/CAA Journal of Automatica Sinica 10.3 (2022): 685-699.

---

> > > > > > > ### Comment · Reviewer_jSAK · 2024-11-29
> > > > > > >
> > > > > > > It is trivial to extend the unconstrained method (https://arxiv.org/pdf/2210.13931) to the constrained setting. Given the bounded domain, one only needs to simplify the proof in that paper to establish the convergence rate of the decentralized Frank-Wolfe method.

---

> > > > > > > > ### Author Response · Authors · 2024-11-29
> > > > > > > >
> > > > > > > > Regarding the decentralized variance-reduction type algorithms, it is indeed common practice to employ similar techniques for bounding the consensus error. In our work, we utilize a Page algorithm like [1], which means certain aspects of our proof may appear familiar. However, there are significant distinctions that set our approach apart.
> > > > > > > >
> > > > > > > > Firstly, existing literature on decentralized unconstrained optimization typically employs a constant step size. In contrast, our paper adopts a diminishing step size of  $\mathcal{O}(\frac{1}{t})$. This choice necessitates a different treatment in the proof, leading us to establish the iterative process $\mathbb{E}[\phi\_{t+1}]\leq \max(1-\frac{p}{2},1-\frac{\eta}{2})\ldots$, contrast to the $\mathbb{E}[\phi_{t+1}]\leq\mathbb{E}[\phi_t-\frac{2}{\eta}\||\nabla f(\bar{x}_t)\||^2-\frac{8m\eta}{3}]$ seen in [1], this adjustment is crucial for addressing the convex case effectively.
> > > > > > > >
> > > > > > > > Secondly, prior research on decentralized Frank-Wolfe algorithms [2, 3, 4] has relied on more intricate proof frameworks to demonstrate convergence. These frameworks, while more complex, have proven to be less efficient compared to our approach. Notably, although earlier work on decentralized unconstrained optimization [5, 6] has existed for some time, the previous decentralized Frank-Wolfe algorithm did not take advantage of them . Therefore, using a more concise and effective proof to achieve better results should be viewed as an advantage, not a disadvantage.
> > > > > > > >
> > > > > > > > References
> > > > > > > >
> > > > > > > > [1] Luo, Luo, and Haishan Ye. "An optimal stochastic algorithm for decentralized nonconvex finite-sum optimization." arXiv preprint arXiv:2210.13931 (2022).
> > > > > > > >
> > > > > > > > [2] X. Jiang, X. Zeng, L. Xie, J. Sun and J. Chen, "Distributed Stochastic Projection-Free Algorithm for Constrained Optimization," in IEEE Transactions on Automatic Control, doi: 10.1109/TAC.2024.3481040.
> > > > > > > >
> > > > > > > > [3] Hou, Jie, et al. "Distributed momentum-based Frank-Wolfe algorithm for stochastic optimization." IEEE/CAA Journal of Automatica Sinica 10.3 (2022): 685-699.
> > > > > > > >
> > > > > > > > [4] Wai, Hoi-To, et al. "Decentralized Frank–Wolfe algorithm for convex and nonconvex problems." IEEE Transactions on Automatic Control 62.11 (2017): 5522-5537.
> > > > > > > >
> > > > > > > > [5] Li, Boyue, Zhize Li, and Yuejie Chi. "DESTRESS: Computation-optimal and communication-efficient decentralized nonconvex finite-sum optimization." SIAM Journal on Mathematics of Data Science 4.3 (2022): 1031-1051.
> > > > > > > >
> > > > > > > > [6] Xin, Ran, Usman A. Khan, and Soummya Kar. "Fast decentralized nonconvex finite-sum optimization with recursive variance reduction." SIAM Journal on Optimization 32.1 (2022): 1-28.

---

> > > > > > > > > ### Comment · Reviewer_jSAK · 2024-11-29
> > > > > > > > >
> > > > > > > > > If you are referring to the learning rate in Theorem 1, it was first introduced in this paper: https://arxiv.org/pdf/2304.11737.

---

> > > > > > > > > > ### Author Response · Authors · 2024-11-30
> > > > > > > > > >
> > > > > > > > > > The learning rate was not proposed by [1], but rather it was introduced by [2] in 2019, which we have clearly stated in the article, with no intention of obscuring this fact. Furthermore, regarding the article you referenced, it seems like a "trivial" extension from unconstrained setting algorithm [3,4].
> > > > > > > > > >
> > > > > > > > > > References
> > > > > > > > > >
> > > > > > > > > > [1] Beznosikov, Aleksandr, David Dobre, and Gauthier Gidel. "Sarah Frank-Wolfe: Methods for Constrained Optimization with Best Rates and Practical Features." Forty-first International Conference on Machine Learning.
> > > > > > > > > >
> > > > > > > > > > [2] Stich, Sebastian U. "Unified optimal analysis of the (stochastic) gradient method." arXiv preprint arXiv:1907.04232 (2019).
> > > > > > > > > >
> > > > > > > > > > [3] Nguyen, Lam M., et al. "SARAH: a novel method for machine learning problems using stochastic recursive gradient." Proceedings of the 34th International Conference on Machine Learning-Volume 70. 2017.
> > > > > > > > > >
> > > > > > > > > > [4] Li, Z., Bao, H., Zhang, X., & Richtárik, P. (2021, July). PAGE: A simple and optimal probabilistic gradient estimator for nonconvex optimization. In International conference on machine learning (pp. 6286-6295). PMLR.

---

### Official Review · Reviewer_CzXL · 2024-11-02

**Soundness:** 3
**Presentation:** 3
**Contribution:** 3
**Rating:** 6
**Confidence:** 4

**Summary:**

The paper studies the decentralized constrained finite-sum optimization problem and provides a projection-free algorithm called DVRGTFW. In the convex and non-convex cases, the sample complexities $\mathcal{O}(n+\sqrt{n/m}L\varepsilon^{-1})$ and $\mathcal{O}(\sqrt{n/m}L^2\varepsilon^{-2})$ are established, respectively. Numerical experiments validate the performance of the algorithm.

**Strengths:**

The paper shows better theoretical convergence results compared to previous works. Specifically, by incorporating techniques such as gradient tracking and multi-consensus, it extends constrained finite-sum algorithms to the decentralized setting. The convergence of DVRGTFW is analyzed using Lyapunov functions, theoretically establishing improved sample and communication complexities, which is also validated by numerical experiments.

**Weaknesses:**

While improved theoretical results are established for decentralized Frank-Wolfe method, the techniques are overall similar to existing ones.

**Questions:**

1. Should the sample complexity in the non-convex case be $\mathcal{O}(n + \sqrt{n/m}L^2\varepsilon^{-2})$? Letting $m = 1$, the problem reduces to the centralized finite-sum setting, where the sample complexity should be $\mathcal{O}(n + \sqrt{n}\varepsilon^{-2})$ or $\mathcal{O}(n\varepsilon^{-2})$, as shown in [1].

2. In Table 1, is a direct comparison of convergence rates with [2] appropriate? Specifically, this paper addresses a finite-sum problem, whereas [2] deals with an online setting. Since DVRGTFW cannot be directly applied to the online problem, such a comparison may be inappropriate. The authors should at least point out the differences in settings when making these comparisons.

3. Finally, there are some minor issues, such as typos.
- The Lyapunov functions defined in L.739 use the symbols  $\Phi$  and  $\Psi$ , but in several places in the following proofs, they are written as  $\phi$  and  $\psi$  (L.994, L.1069, L.1076, L.1082, and L.1085).
- L.818. ``fastMix'' should be ``FastMix''.
- The paper [1] has been accepted in ICML and the reference should be updated.

---
References

[1] Aleksandr Beznosikov, David Dobre, and Gauthier Gidel. Sarah frank-wolfe: Methods for constrained optimization with best rates and practical features. In ICML, 2024.

[2] Hoang Huy Nguyen, Yan Li, and Tuo Zhao. Stochastic constrained decentralized optimization for machine learning with fewer data oracles: a gradient sliding approach. arXiv preprint arXiv:2404.02511, 2024.

---

### Note · Authors · 2024-12-04

I have read and agree with the venue's withdrawal policy on behalf of myself and my co-authors.